# DiffGAD: A Diffusion-based Unsupervised Graph Anomaly Detector

**Jinghan Li[1], Yuan Gao[1†], Jinda Lu[1], Junfeng Fang[1], Congcong Wen[2], Hui Lin[3†], Xiang Wang[1]**

[1]University of Science and Technology of China, [2]New York University,
[3]China Academy of Electronics and Information Technology

```
{lijh111, lujd, fjf}@mail.ustc.edu.cn
{426.yuan, xiangwang1223}@gmail.com
linhui@whu.edu.cn, wencc@nyu.edu
```
[†]Corresponding author.

## ABSTRACT

Graph Anomaly Detection (GAD) is crucial for identifying abnormal entities within networks, garnering significant attention across various fields. Traditional unsupervised methods, which decode encoded latent representations of unlabeled data with a reconstruction focus, often fail to capture critical discriminative content, leading to suboptimal anomaly detection. To address these challenges, we present a *Diffusion-based Graph Anomaly Detector* (DiffGAD). At the heart of DiffGAD is a novel latent space learning paradigm, meticulously designed to enhance the model's proficiency by guiding it with discriminative content. This innovative approach leverages diffusion sampling to infuse the latent space with discriminative content and introduces a content-preservation mechanism that retains valuable information across different scales, significantly improving the model's adeptness at identifying anomalies with limited time and space complexity. Our comprehensive evaluation of DiffGAD, conducted on six real-world and large-scale datasets with various metrics, demonstrated its exceptional performance. Our code is available at: https://github.com/fortunato-all/DiffGAD.

## 1 INTRODUCTION

Graph structure has garnered significant attention from both academia (Gao et al., 2023c; Roy et al., 2023; Gao et al., 2023a; Fang et al., 2024b) and industry (Breuer et al., 2020; Li et al., 2023; Yin et al., 2024), with its potential to represent relationships and structures. Among its wide applications, graph anomaly detection (GAD) has evolved as a popular research topic, aiming to detect abnormal targets (nodes, edges, subgraphs, et al.) from the normal.

Existing research can be categorized into two branches: the semi-supervised learning branch (Dou et al., 2020; Gao et al., 2023b; 2024b) and the unsupervised learning branch (Hamilton et al., 2017; Chen et al., 2020; Peng et al., 2018). Semi-supervised learning approaches utilize a small subset of labeled data to discern patterns of anomalies, enabling the prediction of labels for the remaining unlabeled dataset. However, human annotation is often time-consuming and labor-intensive, which limits the application of semi-supervised methods.

As alternative, unsupervised strategies (Ding et al., 2019; Fan et al., 2020; Sakurada & Yairi, 2014) directly capture node characteristics and local structures without the need for annotation. Specifically, these approaches are on the assumption that *anomalous entities exhibit more complex distributions and are significantly more difficult to reconstruct* (Ding et al., 2019; Fan et al., 2020; Sakurada & Yairi, 2014). Consequently, they utilize Autoencoder (AE) to first map the graph data into latent embeddings (Kingma & Welling, 2014; Kipf & Welling, 2017a; Velickovic et al., 2018) and then decoding, wherein those exhibiting high reconstruction errors are deemed more likely to be anomalies. However, these methods grapple with limitations in their discriminative capability, which results in suboptimal performance. As outlined in (Dou et al., 2020; Liu et al., 2020), abnormal users often deliberately camouflage themselves among normal users. Consequently, they frequently exhibit a significant overlap in common attributes (*e.g.* age, occupation, length and frequency of reviews, etc.) with normal users. This behavior enables these anomalies to divert the focus of

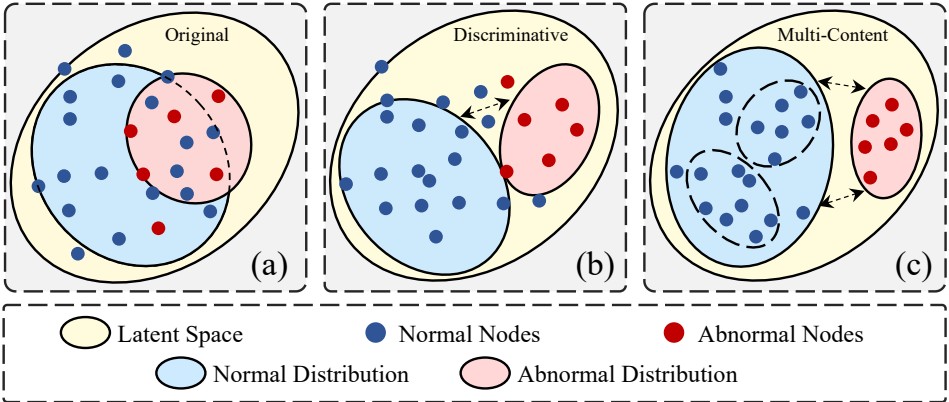

Figure 1: Given several normal and abnormal nodes, the data space is constructed by different methods. Specifically, (a) represents the latent space constructed by current reconstruction-based methods, (b) denotes the latent space learned by our discriminative guidance, (c) is the latent space constructed by introducing the preserved general content on (b).

reconstruction-based models towards overlapping common content, reducing the models' ability to discern truly discriminative features (such as transaction information for fraud detection (Dou et al., 2019)). As a result, anomalous entities are difficult to identify by reconstruction error with such a trivial framework. As illustrated in Figure 1 (a), all data points in the latent space are equally distributed in the learned latent space, without preserving sufficient discriminative content, leading to the overlapping distribution space.

Furthermore, we find that the latent space constructed by the AE-based method (Ding et al., 2019) tends to represent all samples for the Books dataset (Sánchez et al., 2013) into the same point, which is misguided by the huge common content. In addition to the aforementioned issue, the well-known Variational Autoencoder (VAE) based method (Kipf & Welling, 2016) faces a notable challenge: it constructs the latent space within a constrained distribution (e.g., the Gaussian distribution), leading to a uniform latent distribution. This limits the model's expressive capacity, consequently making it challenging to distinguish between abnormal and normal samples. To tackle this issue, we propose to enhance the latent space learning process by incorporating discriminative content. Drawing inspiration from the potent generative capabilities of Diffusion Models (DMs) (Ho et al., 2020; Rombach et al., 2022; Nichol et al., 2022; Dhariwal & Nichol, 2021), we introduce a diffusion-based detection approach, termed DiffGAD. Specifically, our DiffGAD tackles the issue from two aspects:

**Discriminative Distillation to distill discriminative content (Section 3.5).**
Discriminative content is hard to capture due to the sparsity of anomalies, however, shared content is much easier to acquire. Hence we implicitly capture this discriminative information: Initially, we train an unconditional DM to focus on constructing features that encapsulate both discriminative and common elements, referred to as *general content*. Subsequently, a second DM is trained by conditioning on a common feature to learn the latent distribution inclusive of *common content*, where the common feature is constructed by adaptively filtering out potential anomalies in the unconditional DM reconstructed space. By differentiating the general content from this commonality, we isolate the discriminative content. This content is then integrated into the latent space through the application of classifier-free guidance across both trained DMs. As illustrated in Figure 1 (b), this process allows the latent space to segregate normal and abnormal distributions within overlapping areas, thereby enabling the differentiation between abnormal and normal samples.

**General Content Preservation to preserve different scale general content (Section 3.3).**
Given that anomaly detection based on reconstruction error fundamentally aligns more with classification than generation, we introduce slight modifications to the diffusion model to better suit this task. Specifically, during the sampling stage, rather than initiating from a Gaussian distribution, we introduce minor corruptions to the given sample $x$ by adding noises for a small timestep $t < T$, and subsequently denoise it to generate a reconstructed sample $\hat{x}$. Throughout this process, we argue that the general content could be preserved across different scales (reflecting by timestep $t$) to further bolster the model's discriminative power with confidence. In detail, noise at smaller $t$ values preserves large-scale general content, whereas larger $t$ values maintain more localized general

content. As illustrated in Figure 1 (c), by preserving this content, the constructed latent space becomes adept at accurately representing data points. This enhancement effectively widens the distributional discrepancy between anomalous and normal entities, resulting in a robust and discriminative latent space that fosters confident representations.

To summarize, we have the following contributions:

- To the best of our knowledge, we make the first attempt to transfer the diffusion model (DM) from the generation task to a detector in the GAD task and propose a DM-based Graph Anomaly Detector, namely DiffGAD.

- Our DiffGAD enhances the discriminative ability from two aspects: a discriminative content-guided generation paradigm to distill the discriminative content in latent space; and a content-preservation strategy to enhance the confidence of the guidance process.

- Extensive experiments over six real-world and large-scale datasets demonstrate our effectiveness, Theoretical and Empirical computational analysis illustrate our efficiency.

## 2 BACKGROUND

In this section, we revisit some background. Specifically, we first introduce the preliminaries of the Graph Anomaly Detection (GAD) task and then describe the diffusion model in latent space.

### 2.1 TASK FORMULATION

In this work, we focus on unsupervised node-level GAD over static attributed graphs, and each node is associated with an anomaly score, where top-ranked nodes (with large scores) are always indicated as anomalies. In reconstruction-based methods, the anomaly score is reconstruction error. Meanwhile, the data structure can be formalized as an attribute graph $\mathbf{G} = \{\mathcal{V}, \mathbf{X}, \mathcal{E}, \mathbf{A}\} \in \mathcal{G}$, where $\mathcal{V}, \mathbf{X}, \mathcal{E}, \mathbf{A}$ denotes nodes, node attributes, edges, and adjacency matrix, specifically.

### 2.2 DIFFUSION MODEL IN LATENT SPACE

The Latent Diffusion Model (Latent DM) is composed of a pair of forward and reverse diffusion processes based on the unified formulation of Stochastic Differential Equation (SDE) (Song et al., 2021). Concretely, given a latent variable $\boldsymbol{z}$, the SDE can be formalized as:

$$\mathrm{d}\boldsymbol{z} = \boldsymbol{f}(\boldsymbol{z}, t)\mathrm{d}t + g(t)\mathrm{d}\boldsymbol{w}_t, \tag{1}$$

where $\boldsymbol{f}(\cdot)$ and $g(\cdot)$ are the drift and diffusion functions, respectively, and $\boldsymbol{f}(\cdot)$ can also be expressed in the form of $\boldsymbol{f}(\boldsymbol{z}, t) = \boldsymbol{f}(t)\boldsymbol{z}$. Then we define the forward, backward, and training process as:

**Forward Process.** Given the latent variable $\boldsymbol{z}$, the forward process transforms $\boldsymbol{z}$ with noises to construct a sequence of step-dependent variables $\{\boldsymbol{z}_t\}_{t=0}^T$, where $\boldsymbol{z}_0 = \boldsymbol{z}$ is the initial point of this process. Specifically, with SDE, the diffusion kernel is denoted as a conditional distribution of $\boldsymbol{z}_t$:

$$p\left(\boldsymbol{z}_t \mid \boldsymbol{z}_0\right) = \mathcal{N}(s(t)\boldsymbol{z}_0, s^2(t)\sigma^2(t)\mathbf{I}), \tag{2}$$

where $s(t)$ and $\sigma(t)$ are step-dependent to control the noise level. We follow an efficient design (Karras et al., 2022; Zhang et al., 2023a) to simplify the diffusion kernel, where we set $s(t) = 1$ and $\sigma(t) = t$ in this work. Therefore, the forward process can be formulated as:

$$\boldsymbol{z}_t = \boldsymbol{z}_0 + \sigma(t)\varepsilon, \text{where } \varepsilon \sim \mathcal{N}\left(\mathbf{0}, \mathbf{I}\right). \tag{3}$$

**Reverse Process.** The reverse process reconstructs the latent variable $\boldsymbol{z}$ by predicting and removing the added noises. Specifically, with SDE and our diffusion kernel simplification (Karras et al., 2022), the reverse process can be derived as:

$$\mathrm{d}\boldsymbol{z} = -2\dot{\sigma}(t)\sigma(t)\nabla_{\boldsymbol{z}} \log p_t(\boldsymbol{z})\mathrm{d}t + \sqrt{2\dot{\sigma}(t)\sigma(t)}\mathrm{d}\boldsymbol{\omega}_t, \tag{4}$$

where $\nabla_{\boldsymbol{z}} \log p_t(\boldsymbol{z})$ is the score function of $\boldsymbol{z}$, and $\dot{\sigma}(t)$ is the first order derivative of $\sigma(t)$.

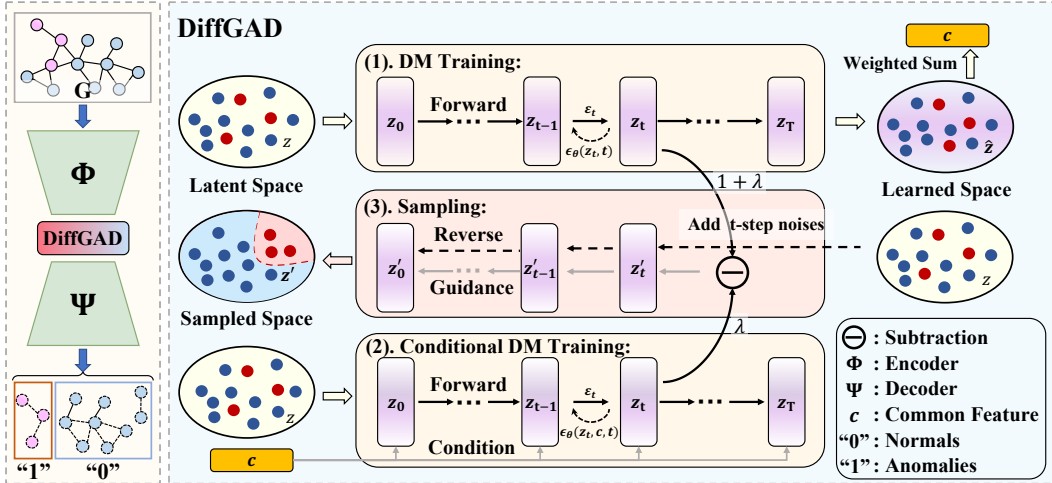

Figure 2: An overview of DiffGAD. Given a graph, we first encode it into latent space, and we then reconstruct it with both unconditioned and conditioned diffusion models to distill the discriminative content. Finally, we decode the reconstructed latent embedding for anomaly detection.

**Training Process.** The training process matches the noises in the forward and reverse processes, which can be achieved by denoising score matching (Song et al., 2021) as:

$$\mathcal{L}_{\text{DM}} = \mathbb{E}_{z_0 \sim p(z)} \mathbb{E}_{z_t \sim p(z_t | z_0)} \| \epsilon_\theta (z_t, t) - \varepsilon \|_2^2, \tag{5}$$

where $\epsilon_\theta$ is the neural network. After training, the diffusion model can be applied for further generation by the reverse process with the score function as $\nabla_z \log p_t(z) = -\epsilon_\theta / \sigma(t)$ in Eq. 4.

## 3 METHODOLOGY

In this section, we illustrate DiffGAD in detail.

### 3.1 OVERVIEW

Figure 2 overviews our proposed DiffGAD, which functions primarily within the latent space. Upon receiving a graph, our methodology initiates by mapping it into the latent space, as detailed in §3.2. Within this transformed domain, we employ a dual Diffusion Model (DM) approach. The first DM is tasked with encapsulating the general content, as elaborated in §3.3, while the second DM targets the extraction of common content, described in §3.4. Herein, the discriminative content is identified by the disparities between these two DM-captured representations. Inspired by (Ho & Salimans, 2022), the concurrent sampling of the two DMs can be enabled by controlling a simple hyper-parameter. It is worth noting that our model's foundation is rooted in a well-established hypothesis, supported by works such as (Ding et al., 2019; Fan et al., 2020; Sakurada & Yairi, 2014), which posits that anomalous entities exhibit more complex distributions and are significantly more difficult to reconstruct. Consequently, nodes with higher reconstruction error are more likely to be anomalies. To facilitate this detection mechanism, our framework requires the original samples with their respective reconstructed counterparts, a process systematically described in §3.3.

### 3.2 LATENT SPACE PROJECTION

In this work, we leverage an encoder to map graph features into a latent representation space, a pivotal step that precedes the detailed exposition of diffusion models in our work. Thus, it is essential to first delineate the methodology employed in projecting graph characteristics into this latent space.

More formally, for a given graph **G**, we utilize the Graph Autoencoder (AE) framework as delineated in (Ding et al., 2019) to facilitate its transformation into the latent space. The architecture of the Graph AE comprises two primary components: an encoder, denoted as $\Phi$, and a decoder, denoted

---

**Algorithm 1** The training and inference procedure of DiffGAD.

---

**Input:** An attribute graph $\mathbf{G} = \{\mathcal{V}, \mathbf{X}, \mathcal{E}, \mathbf{A}\}$,
**Output:** The detection scores of each node in $\mathbf{G}$.
1:  Train Graph AE $\{\mathbf{\Phi}, \mathbf{\Psi}\}$ by Eq. 12.
2:  Extract latent embedding $\boldsymbol{z}$ by Eq. 6.
3:  Initialize common feature $c_0$ as the mean of $\boldsymbol{z}$ and set $c_{\text{current}} = c_0$.
4:  **while** Training unconditional DM $\boldsymbol{\epsilon}_\theta(\boldsymbol{z}_t, t)$ **do**
5:      Update $\boldsymbol{\epsilon}_\theta(\boldsymbol{z}_t, t)$ by Eq. 5,
6:      Update common feature $c_{\text{next}}$ by Eq. 9 and Eq. 10,
7:      Set $c_{\text{current}} = c_{\text{next}}$.
8:  **end while**
9:  Set $c = c_{\text{next}}$,
10: Train conditional DM $\boldsymbol{\epsilon}_\theta(\boldsymbol{z}_t, c, t)$ by Eq. 13.
11: Add t-step noises to latent embedding $\boldsymbol{z}$ by Eq. 3.
12: Reconstruct the noisy embedding with the modified score in Eq. 11.
13: Decode the reconstructed embedding by Eq. 7.
14: Calculate detection scores between $\mathbf{G}$ and the decoded graph by Eq. 12.

---

as $\mathbf{\Psi}$. The encoder function, $\mathbf{\Phi}$, is designed to process the node feature matrix $\mathbf{X} \in \mathbb{R}^{n \times d}$ and the adjacency matrix $\mathbf{A} \in \mathbb{R}^{n \times n}$, thereby yielding latent feature embedding $\boldsymbol{z}$, computed as:

$$\boldsymbol{z} = \mathbf{\Phi}(\mathbf{X}, \mathbf{A}), \tag{6}$$

wherein $\mathbf{\Phi}$ incorporates a Graph Convolutional Network (GCN) (Kipf & Welling, 2017a) as its underlying mechanism for aggregating information from node features $\boldsymbol{X}$ into latent embedding $\boldsymbol{z}$, with $\boldsymbol{z} \in \mathbb{R}^{n \times k}$ and $k$ representing the dimensionality of the latent space.

Subsequently, the decoder component is tasked with the reconstruction of the node feature matrix and adjacency matrix from the latent embedding $\boldsymbol{z}$, expressed through the equations:

$$\begin{cases} \hat{\mathbf{X}} = \mathbf{\Psi}_{\text{feat}}(\boldsymbol{z}, \mathbf{A}), \\ \hat{\mathbf{A}} = \mathbf{\Psi}_{\text{stru}}(\boldsymbol{z}\boldsymbol{z}^T), \end{cases} \tag{7}$$

in which $\mathbf{\Psi}_{\text{feat}}$ and $\mathbf{\Psi}_{\text{stru}}$ are specifically designed for the prediction of node features and graph structure (edges), respectively. This meticulous elaboration of the latent space projection methodology set the stage for a comprehensive understanding of our novel DiffGAD framework.

### 3.3 GENERAL CONTENT PRESERVATION

Once the latent space is ready, the reconstruction error by DM acts as a proxy of the anomaly score associated with individual nodes. To achieve this, pairs of original and reconstructed samples are necessary (Gao et al., 2024a). Our general content preservation aims to adapt the DM architecture to the reconstruction error-based anomaly detection task. Unlike traditional DMs, which samples random Gaussian noise to generate $\hat{\boldsymbol{z}}_0$, our approach perturbs original latent embedding $\boldsymbol{z}_0$ by mixing random noises from different scales as the initial points for sampling. Concretely, following the work in Ho et al. (2020), we categorize the noises into 500 different scales, and we add noises at each scale by the following equation:

$$z_t = \frac{t}{T} z_0 + \frac{T-t}{T} \epsilon \tag{8}$$

where $\epsilon$ is a random noise, and $\epsilon \in \mathcal{N}(0, 1)$, this corrupted embedding $z_t$ is then used to generate reconstructed sample $\hat{\boldsymbol{z}}_0$ with the aim of preserving the general content. The benefit of this process are threefolds: (1) It facilitates the computation of reconstruction error by aligning original and reconstructed sample; (2) By modulating $t$, the extent of preserved general content can be controlled; a lower $t$ value results in a reconstruction that is closer to the original content, thereby retaining more of the general content. (3) As the DM functions without explicit conditioning, it inherently associates with the general content, which lays the groundwork for the introduction of discriminative content.

### 3.4 COMMON FEATURE CONSTRUCTION

In this study, we treat divergence of unconditional and conditional DMs as the discriminative content of interest. Besides unconditional DM, our approach also necessitates the incorporation of a conditional DM. This subsection is dedicated to elucidating the conditioning mechanism construction.

Consider the latent embedding $z = \{z^v\}_{v=1}^n$, where each $z^v \in \mathbb{R}^k$ denotes the embedding of the $v$-th node. We initialize the common feature $c_0$ as the mean value over all latent node embeddings, capturing a global perspective of the node distribution and setting a prior for their generative process. This feature acts as the conditioning and is not fixed during the training of DM, next we demonstrate the adaptive refinement of this feature in depth.

During the training phase of the unconditional DM, which we represent as $\epsilon_\theta(z_t, t)$, the conditioning undergoes iterative updates. To illustrate, we identify the common feature conditioning at any training iteration as $c_{\text{current}}$. The feature update is governed by:

$$c_{\text{next}} = \sum_{v \in \mathcal{V}} \omega_v \cdot \hat{z}^v, \tag{9}$$

where $c_{\text{next}}$ is the updated common feature for the next training iteration, and $\omega_v$ is the weighting vector to control the update process. We obtain $\omega_v$ by similarity calculation between the reconstructed node embedding $\hat{z}^v$ and current common feature $c_{\text{current}}$, which can be formalized as:

$$\omega_v = \frac{exp(\bar{\omega}_v/\tau)}{\sum exp(\bar{\omega}_v/\tau)}, \text{where } \bar{\omega}_v = \cos\langle \hat{z}^v, c_{\text{current}} \rangle, \tag{10}$$

where $\tau$ is the temperature parameter to control the smoothness of weights, $\cos\langle \cdot, \cdot \rangle$ denotes the cosine similarity. The update of the common feature involves aggregating the contribution of all nodes, weighted by their respective $\omega_v$ values. A larger $\omega_v$ indicates a more significant contribution of the current $\hat{z}^v$ to the update process. Notably, nodes that significantly deviate from the current common feature, $c_{\text{current}}$, are assigned with lower weights during the update. The underlying rationale is to mitigate the impact of potential anomalies and to derive a comprehensive common feature. Upon completing this iteration, $c_{\text{next}}$ is designated as the new current common feature $c_{\text{current}}$ for the forthcoming update cycle. Furthermore, we establish the final common feature $c$ which corresponds to $c_{\text{next}}$ from the last training iteration. This finalized feature remains static during the detection phase. This approach ensures that the common feature is reflective of the dataset's core attributes, which leads us to the discriminative content calculation.

### 3.5 DISCRIMINATIVE CONTENT DISTILLATION

Leveraging the unconditional DM, $\epsilon_\theta(z_t, t)$, and the common feature $c$, our approach is able to extract the discriminative content through discriminative content distillation. Specifically, we first train a conditional DM as $\epsilon_\theta(z_t, c, t)$, which aims to learn the common knowledge by reconstructing latent embedding $z = \{z^v\}_{v=1}^n$ with common feature $c$ as conditional information.

Inspired by the idea of classifier-free guidance (Ho & Salimans, 2022), we distill the discriminative content by performing a linear combination of the unconditional and conditional DMs. Differently, our unconditional DM $\epsilon_\theta(z_t, t)$ contains both discriminative and common content, and the conditional DM $\epsilon_\theta(z_t, c, t)$ captures the common content, which describes that "what most of the reconstructed nodes focus on". Thus, the discriminative content is achieved by subtracting the general content from the common content learned from the unconditional DM and conditional DM, respectively. Specifically, the modified score can be denoted as:

$$\tilde{\epsilon}_\theta(z_t, c, t) = (1+\lambda)\epsilon_\theta(z_t, t) - \lambda\epsilon_\theta(z_t, c, t),$$
$$\nabla_z \log p_t(z) = -\tilde{\epsilon}_\theta(z_t, c, t)/\sigma(t), \tag{11}$$

where $\lambda$ is the hyperparameter to regulate the strength. By utilizing the modified score for the sampling process, we distill the discriminative content into the latent space, and then obtain the discriminative latent embeddings.

Table 1: The ROC-AUC performance with different components of our method.

| Method | Weibo | Reddit | Disney | Books | Enron | Avg |
|---|---|---|---|---|---|---|
| AE | $92.0 \pm 0.1$ | $56.1 \pm 0.0$ | $40.3 \pm 6.7$ | $59.1 \pm 2.5$ | $59.1 \pm 1.6$ | $61.3 \pm 2.2$ |
| Diff | $91.6 \pm 0.5$ | $55.8 \pm 0.1$ | $49.1 \pm 0.3$ | $58.3 \pm 2.5$ | $57.7 \pm 1.8$ | $62.5 \pm 1.0$ |
| Cond-Diff | $91.4 \pm 0.5$ | $55.9 \pm 0.1$ | $49.2 \pm 0.4$ | $58.8 \pm 1.8$ | $57.6 \pm 1.9$ | $62.6 \pm 0.9$ |
| **DiffGAD** | $\mathbf{93.4} \pm 0.3$ | $\mathbf{56.3} \pm 0.1$ | $\mathbf{54.5} \pm 0.2$ | $\mathbf{66.4} \pm 1.8$ | $\mathbf{71.6} \pm 7.0$ | $\mathbf{68.4} \pm 1.9$ |

## 3.6 TRAINING AND INFERENCE

**Training Stage.** In the training stage, given an attribute graph $\mathbf{G} = \{\mathcal{V}, \mathbf{X}, \mathcal{E}, \mathbf{A}\}$, we firstly train the Graph AE by the following objective:

$$\mathcal{L}_{\text{AE}} = \alpha \cdot ||\mathbf{X} - \hat{\mathbf{X}}||_2 + (1 - \alpha) \cdot ||\mathbf{A} - \hat{\mathbf{A}}||_2, \tag{12}$$

where $|| \cdot ||_2$ denotes the L2 norm, and $\alpha$ is a hyper-parameter to balance the effect of feature and structural reconstruction. Then we train the unconditional DM $\epsilon_\theta (z_t, t)$ by the training objective as in Eq. 5, meanwhile, we obtain the common feature $c$. Finally, we train the conditional DM $\epsilon_\theta (z_t, c, t)$ by the following equation:

$$\mathcal{L}_{\text{Cond}} = \mathbb{E}_{z_0 \sim p(z)} \mathbb{E}_{z_t \sim p(z_t | z_0, c)} \left\| \epsilon_\theta (z_t, c, t) - \varepsilon \right\|_2^2. \tag{13}$$

**Inference Stage.** In the inference stage, given attribute graph $\mathbf{G}$, we first transform it into latent space by well-trained encoder $\Phi$. Second, we add $t$-step noises ($t < T$) on the extracted latent embedding to preserve the general content. Then, we discriminatively sample from both unconditional DM $\epsilon_\theta (z_t, t)$ and conditional DM $\epsilon_\theta (z_t, c, t)$ with Eq. 11 as scores. Finally, we transform the reconstructed embedding into graph space by decoder $\Psi$ for reconstruction error calculation.

## 4 EXPERIMENT

In this section, we conduct experiments to validate the effectiveness of our DiffGAD. Specifically, we first introduce the experimental settings, and next, we analyze the ablation studies, finally, we describe the comparison results with the state-of-the-art methods.

## 4.1 EXPERIMENTAL SETTINGS

**Datasets.** Following the work in (Liu et al., 2022b) we employ 13 baselines as benchmarks on 6 real-world datasets (Weibo (Zhao et al., 2020), Reddit (Kumar et al., 2019; Wang et al., 2021), Disney (Sánchez et al., 2013), Books (Sánchez et al., 2013), Enron (Sánchez et al., 2013)), including a large-scale dataset Dgraph (Huang et al., 2022) for evaluation.
**Metrics.** Following the extensive literature in GAD (Liu et al., 2022b; Ding et al., 2019; Kipf & Welling, 2016), we compressively evaluate the performance of DiffGAD with the representative ROC-AUC (Receiver Operating Characteristic Area Under Curve), AP (Average Precision), Recall@k, and the AUPRC (Area Under the Precision and Recall Curve) metrics.
**Comparisons.** Following the work in (Liu et al., 2022b), we report the average performance with std results over 20 trials for a fair comparison. Moreover, we re-implement current methods for all datasets with (Liu et al., 2022a), and observe a large performance gap for the ROC-AUC in Enron, while other datasets are similar (as the official GitHub issue mentioned in [1]). Therefore, to prevent ambiguity, we use the re-implemented ROC-AUC results on the Enron dataset and follow the results of other datasets and metrics in (Liu et al., 2022b).

## 4.2 ABLATION STUDIES

**The effects of discriminative content distillation.** We conduct experiments over various $\lambda$ in Eq. 11, and the experimental results are shown in Figure 3. Specifically, we list the AUC performance with the conditional DM $\epsilon_\theta (z_t, c, t)$ as $\lambda = -1.0$ (only common content), with the unconditional DM $\epsilon_\theta (z_t, t)$ as $\lambda = 0.0$ (only general content), and with different values of $\lambda$ ranging from 0.2 to 2.0

---
[1]https://github.com/pygod-team/pygod

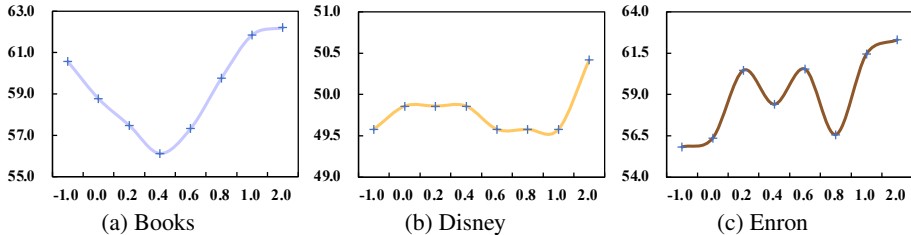

Figure 3: The ROC-AUC performance of 3 representative datasets under the different scale of the control of $\lambda$, where $x$ axis represents the variance of $\lambda$, and $y$ axis is the ROC-AUC results.

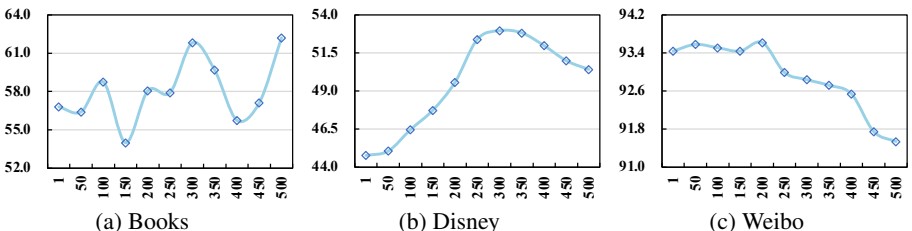

Figure 4: The ROC-AUC performance of different timesteps $t$ over 3 representative datasets, where $x$ axis represents different timesteps $t$, and $y$ axis is the ROC-AUC results.

over the Books, Disney, and Enron datasets. From Figure 3, we find that different datasets have distinct sensitivities of $\lambda$, and the performance on Books and Enron are sensitive to the various $\lambda$. Moreover, we can observe that the best results are achieved with large $\lambda$ ($\lambda = 2.0$) for all 3 datasets, which not only demonstrates the effectiveness of our discriminative content distillation but also shows that the discriminative content is hard to mine for those datasets.

**The influences of general content preservation.** We conduct experiments over various timestep $t$ in Sec. 3.3 to preserve general content from different scales, and the experimental results are illustrated in Figure 4. Specifically, we list the AUC performance with timestep varies from 1 to 500 over the Books, Disney, and Weibo datasets. The full diffusion timestep is 500, where $t = 1$ means sampling from large-scale general content, and $t = 500$ implies sampling from the noise. Specifically, from Figure 4, we can find that (1) general content from different scales is required for different datasets, where Weibo requires more general content (small $t$), and less general content is needed for Books and Disney (large $t$), and (2) compared with sampling from the noises ($t = 500$), our general content preservation improves over all datasets (various $t$), which demonstrates our effectiveness.

**The effects of different components.** We conduct experiments with different components such as the Autoencoder ("AE"), the unconditional DM ("Diff"), and the conditional DM ("Cond-Diff"), the experimental results are shown in Table 1, where "DiffGAD" is our method. Specifically, we can find that: (1) applying the DM over the AE model achieves marginal performance gains ("Diff" vs "AE"), which demonstrates that simply utilizing DM can't tackle the lack of discriminative content issue. (2) the conditional DM and the unconditional DM obtain similar results ("Diff" vs "Cond-Diff"), which proves our hypothesis, that in the general content, the huge common content prioritizes the discriminative. (3) our method attains significant improvements over different components ("DiffGAD" vs others), this proves our effectiveness, which could distill the discriminative

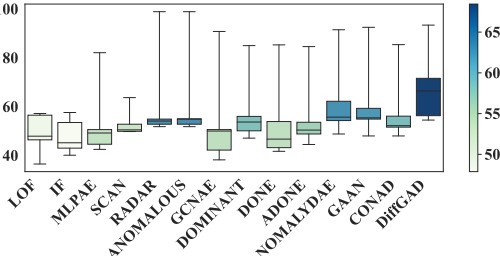

Figure 5: Average ROC-AUC performance over 5 datasets, where the color represents the average AUC, and the central line is the median (Many methods for Dgraph encounter OOM and TLM restriction, thus Dgraph is omitted).

Table 2: Performance comparison (**ROC-AUC**) among 13 algorithms on 6 datasets, where we show the *avg perf.* $\pm$ *the std of perf.* of each method. The best results of all methods are indicated in boldface, and the second best results are underlined. OOM refers to out-of-memory with CPU. TLE denotes time limit of 24 hours exceeded. We list the Avg result over 5 datasets, due to the unavailable Dgraph results with OOM and TLE for many methods, we omit Dgraph.

| Algorithm | Weibo | Reddit | Disney | Books | Enron | Avg | Dgraph |
|---|---|---|---|---|---|---|---|
| *graph-agnostic* | | | | | | | |
| LOF | $56.5 \pm 0.0$ | $\mathbf{57.2 \pm 0.0}$ | $47.9 \pm 0.0$ | $36.5 \pm 0.0$ | $46.4 \pm 0.0$ | $48.9 \pm 0.0$ | TLE |
| IF | $53.5 \pm 2.8$ | $45.2 \pm 1.7$ | $\mathbf{57.6 \pm 2.9}$ | $43.0 \pm 1.8$ | $40.1 \pm 1.4$ | $47.9 \pm 2.1$ | $\mathbf{60.9 \pm 0.7}$ |
| MLPAE | $82.1 \pm 3.6$ | $50.6 \pm 0.0$ | $49.2 \pm 5.7$ | $42.5 \pm 5.6$ | $44.6 \pm 7.1$ | $53.8 \pm 4.4$ | $37.0 \pm 1.9$ |
| *classical algorithms* | | | | | | | |
| SCAN | $63.7 \pm 5.6$ | $49.9 \pm 0.3$ | $50.5 \pm 4.0$ | $49.8 \pm 1.7$ | $52.8 \pm 3.4$ | $53.3 \pm 3.0$ | TLE |
| Radar | $\mathbf{98.9 \pm 0.1}$ | $54.9 \pm 1.2$ | $51.8 \pm 0.0$ | $52.8 \pm 0.0$ | $54.1 \pm 10.1$ | $62.5 \pm 2.3$ | OOM |
| ANOMALOUS | $\mathbf{98.9 \pm 0.1}$ | $54.9 \pm 5.6$ | $51.8 \pm 0.0$ | $52.8 \pm 0.0$ | $55.0 \pm 9.8$ | $\underline{62.7 \pm 3.1}$ | OOM |
| *deep algorithms* | | | | | | | |
| GCNAE | $90.8 \pm 1.2$ | $50.6 \pm 0.0$ | $42.2 \pm 7.9$ | $50.0 \pm 4.5$ | $38.2 \pm 6.5$ | $54.4 \pm 4.0$ | $40.9 \pm 0.5$ |
| DOMINANT | $85.0 \pm 14.6$ | $56.0 \pm 0.2$ | $47.1 \pm 4.5$ | $50.1 \pm 5.0$ | $53.7 \pm 4.2$ | $58.4 \pm 5.7$ | OOM |
| DONE | $85.3 \pm 4.1$ | $53.9 \pm 2.9$ | $41.7 \pm 6.2$ | $43.2 \pm 4.0$ | $46.7 \pm 6.1$ | $54.2 \pm 4.7$ | OOM |
| AdONE | $84.6 \pm 2.2$ | $50.4 \pm 4.5$ | $48.8 \pm 5.1$ | $53.6 \pm 2.0$ | $44.5 \pm 2.9$ | $56.4 \pm 3.3$ | OOM |
| AnomalyDAE | $91.5 \pm 1.2$ | $55.7 \pm 0.4$ | $48.8 \pm 2.2$ | $\underline{62.2 \pm 8.1}$ | $54.3 \pm 11.2$ | $62.5 \pm 4.6$ | OOM |
| GAAN | $92.5 \pm 0.0$ | $55.4 \pm 0.4$ | $48.0 \pm 0.0$ | $54.9 \pm 5.0$ | $\underline{59.3 \pm 0.2}$ | $62.0 \pm 1.1$ | OOM |
| CONAD | $85.4 \pm 14.3$ | $56.1 \pm 0.1$ | $48.0 \pm 3.5$ | $52.2 \pm 6.9$ | $51.6 \pm 4.3$ | $58.7 \pm 5.8$ | $34.7 \pm 1.2$ |
| **DiffGAD** | $\underline{93.4 \pm 0.3}$ | $\underline{56.3 \pm 0.1}$ | $\underline{54.5 \pm 0.2}$ | $\mathbf{66.4 \pm 1.8}$ | $\mathbf{71.6 \pm 7.0}$ | $\mathbf{68.4 \pm 1.9}$ | $\underline{52.4 \pm 0.0}$ |

content and further boost the detection performance. Moreover, we also visualize the reconstructed distribution by different components in Appendix B.

## 4.3 COMPARISON WITH SOTA METHODS.

In this subsection, we comprehensively compare our method with various state-of-the-art algorithms, ranging from graph-agnostic algorithms LOF (Breunig et al., 2000), MLPAE (Sakurada & Yairi, 2014), IF (Liu et al., 2012), and classical algorithms Rador (Li et al., 2017), ANOMALOUS (Peng et al., 2018), SCAN (Xu et al., 2007), to deep algorithms GAAN (Chen et al., 2020), DOMINANT (Ding et al., 2019), GCNAE (Kipf & Welling, 2016), DONE (Bandyopadhyay et al., 2020), CONAD (Xu et al., 2022), and the comparison results are reported in Table 2. Moreover, we also show the average of AUC results with standard deviation ( denoted as "Avg") in the table, and then present a box chart to show the average AUC results of 5 datasets in Figure 5.

Specifically, we draw the following observations: **(1)** DiffGAD achieves "an outlier node detection method that works universally well on all datasets". It obtains top-2 AUC results over all datasets and significantly outperforms current methods with a large margin in average AUC with more than 9% average AUC (compared with the second Average AUC results in ANOMALOUS). **(2)** DiffGAD demonstrates robustness and stability across different datasets, it attains small *std* results over different datasets. **(3)** DiffGAD outperforms other deep-based algorithms by a large margin. Specially, the best improvement is for the Enron dataset, compared with the GAAN method, our DiffGAD attains more than 20.7% AUC gains. These achieved performances are significant, and this can be attributed to the enhancement of discriminative ability. More detailed analysis is in Appendix C.1.

## 5 TIME AND COMPUTATIONAL ANALYSIS

In this study, we analyze the time and computational analysis from theoretical and empirical views.

### 5.1 THEORETICAL ANALYSIS

**(1)** Graph AutoEncoder. We analyze the complexity according to (Liu et al., 2022b). Specifically, we utilize the Graph convolutional network as our backbone, whose complexity is linear to the edge numbers. For each layer, the convolution operation is $\tilde{D}^{-\frac{1}{2}} \tilde{A} \tilde{D}^{-\frac{1}{2}} X W$, and thus the complexity is $\mathcal{O}(mdh)$ (Kipf & Welling, 2017b), where $\tilde{A} X$ can be implemented efficiently with sparse-dense matrix multiplication. For $\mathcal{O}(mdh)$, $m$ is the number of non-zero elements in matrix $A$, $d$ is the feature dimensions for the attributed network, and $h$ is the number of feature maps of the weight

Table 3: **Wall-clock running time (s)** among deep algorithms on five different numbers of epochs, where the experiments are conducted on the *gen_time* dataset by BOND (Liu et al., 2022b).

| Algorithm | 10 | 100 | 200 | 300 | 400 |
|---|---|---|---|---|---|
| GCNAE | 0.27 | 2.40 | 4.46 | 6.80 | 10.95 |
| DOMINANT | 0.10 | 0.59 | 1.35 | 1.90 | 2.40 |
| DONE | 0.11 | 0.75 | 1.61 | 2.26 | 3.18 |
| AdONE | 0.14 | 1.11 | 2.55 | 4.12 | 5.30 |
| AnomalyDAE | 0.11 | 0.70 | 1.12 | 1.71 | 2.04 |
| GAAN | 0.09 | 0.61 | 1.17 | 1.44 | 1.89 |
| CONAD | 0.17 | 1.23 | 2.40 | 3.49 | 4.59 |
| **DiffGAD (AE $\alpha \neq 1$)** | 0.07 | 0.65 | 1.26 | 1.91 | 2.55 |
| **DiffGAD (AE $\alpha = 1$)** | 0.07 | 0.51 | 0.97 | 1.48 | 1.96 |
| **DiffGAD(Diff)** | 0.03 | 0.42 | 0.81 | 1.21 | 1.63 |
| **DiffGAD(Cond-Diff)** | 0.03 | 0.45 | 0.91 | 1.35 | 1.80 |
| **DiffGAD(Sample)** | | | 0.17 | | |

Table 4: **GPU Memory Consumption (MB)** among deep algorithms on five different graph sizes (nodes), where the experiments are conducted on the *gen* dataset by BOND (Liu et al., 2022b).

| Algorithm | 100 | 500 | 1000 | 5000 | 10000 |
|---|---|---|---|---|---|
| GCNAE | 180 | 200 | 200 | 200 | 262 |
| DOMINANT | 218 | 224 | 240 | 750 | 2194 |
| DONE | 218 | 226 | 264 | 900 | 2610 |
| AdONE | 220 | 260 | 292 | 904 | 2592 |
| AnomalyDAE | 218 | 228 | 262 | 1032 | 3756 |
| GAAN | 220 | 228 | 270 | 1032 | 3358 |
| CONAD | 218 | 226 | 246 | 864 | 2578 |
| **DiffGAD** | 220 | 228 | 246 | 830 | 2532 |

matrix. Moreover, to capture graph topological content, we devise a link prediction layer to reconstruct the original topological structure, and thus the overall complexity is $\mathcal{O}(mdH + n^2)$, where $H$ is the summation of all feature maps across different layers, and $n$ is the number of nodes.

(**2**) Latent Diffusion Models. We employ an MLP as the denoising function by following (Karras et al., 2022; Zhang et al., 2023a). Specifically, during the training phase, a random timestep is sampled for each epoch to train the latent node embeddings. The time complexity of this process is $\mathcal{O}(end'H')$, where $e$ denotes the number of training epochs, $H'$ is the number of feature maps for MLP, $d'$ is the input dimension for DM, which is also the feature map dimension of AE, since we adopt the output of AE as the input of DM. During the inference phase, similarly, the time complexity is $\mathcal{O}(tnd'H')$, where $t$ represents the number of sampling timesteps in the denoising process. The combined time complexity of both training and inference phases remains $\mathcal{O}((e + t)nd'H')$.

## 5.2 EMPIRICAL DISCUSSION

Following (Liu et al., 2022b), we employ the Wall-Clock time and GPU memory for empirical comparisons, and the experimental results are listed in Table 3 and Table 4, respectively. The whole testing is conducted on a Linux server with a 2.90GHz Intel(R) Xeon(R) Platinum 8268 CPU, 1T RAM, and 1 Nvidia 2080 Ti GPU with 11GB memory.

We can observe that DiffGAD is efficient in terms of both time and memory usage. Specifically, benefitting from the advancements in diffusion acceleration, we adopt a more efficient EDM sampler (Karras et al., 2022) within latent space and utilize 50 sampling steps. From Table 4, we can observe that the sampling time of DiffGAD is quite short, only 0.17 seconds. Moreover, we also list the Autoencoder with $\alpha = 1$ in Eq. 7, Denoted as **DiffGAD (AE $\alpha = 1$)**, which omits the topology reconstruction, and reduces AE costs to $\mathcal{O}(mdH)$. We can observe that the running time of AE $\alpha = 1$ and diffusion model are comparable.

Furthermore, as the size of the graph scales to real-world proportions, the overall time complexity of DM simplifies to $\mathcal{O}(n)$ as we discuss in 5.1. This indicates that the complexity $\mathcal{O}(n^2)$ of AutoEncoder dominates the whole complexity, while the time required for diffusion model who adding and removing noise in latent space can be neglected.

## 6 CONCLUSION

In this work, we make the first effort to transfer the diffusion model from generation task to a detector and propose a diffusion-based unsupervised graph anomaly detector, namely DiffGAD. Specifically, (1) a discriminative content-guided generation paradigm is proposed to capture the discriminative content and then distill it into the latent space, and (2) a content-preservation strategy is designed to enhance the confidence of the aforementioned guidance process, and (3) extensive experiments on 6 real-world and large-scale datasets with different metrics demonstrate the effectiveness of our method, and (4) comprehensive time and computational analysis demonstrate our efficiency.

**Limitation and Future Work.** The expressiveness of latent embeddings might be limited by graph encoder, we will explore some encoder-free strategies in the future.

## 7 CODE OF ETHICS AND REPRODUCIBILITY STATEMENT

**Code of Ethics.** We do not foresee any direct, immediate, or negative societal impacts stemming from the outcomes of our research. **Reproducibility Statement.** All results presented in this work are fully reproducible. We provide our code using the GitHub link. The optimal hyperparameters are detailed in Appendix D.

## 8 ACKNOWLEDGEMENTS

This research is supported by the National Natural Science Foundation of China (92270114). This research was also supported by the advanced computing resources provided by the Supercomputing Center of the USTC.

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

## A    RELATED WORKS

In this section, we introduce notable prior works on graph anomaly detection, list the related diffusion model-based methods, and enumerate the differences between DiffGAD and related methods.

**Graph Anomaly Detection (GAD)** aims to detect abnormal targets from a huge number of normal samples. In the semi-supervised branch, Graph Neural Networks (GNNs) (Wang et al., 2019; He et al., 2024; Zhang et al., 2024) have achieved remarkable success in detecting graph anomalies across various domains for its excellent representation capabilities for modeling complex relationships. Specially, CapsGI (Zheng et al., 2024) combines capsules with self-supervised learning (SSL) to overcome the inconsistency of anomaly detection on graphs. PMP (Zhuo et al., 2024) introduces Partitioning Message Passing to adaptively adjust the information aggregated from its heterophilic and homophilic neighbors. RQGNN (Dong et al., 2024) proves that the accumulated spectral energy of the graph signal can be represented by its Rayleigh Quotient, and proposes Rayleigh Quotient GNN for graph-level anomaly detection.

To tackle the issue of limited labeled data, recent methods (Fan et al., 2020; Yuan et al., 2021; Xu et al., 2022) propose unsupervised approaches over the large-scale unlabeled data, where the anomaly score is calculated as the reconstruction error. For example, the representative work in (Ding et al., 2019) uses GCN (Kipf & Welling, 2017a) and AE (Kingma & Welling, 2014) to encode the graph into a high-dimensional latent space, decodes the structure and attributes separately using the decoder, and then utilizes the weighted reconstruction error of node feature and structure as the anomaly score.

**Diffusion Models (DMs)** have garnered widespread attention for their remarkable advances in generating high-quality images and videos. Fueled by the explosion of deep learning (Lu et al., 2025; Fang et al., 2024a; Lu et al., 2023; 2024), representative models like DDPM (Ho et al., 2020; Sohl-Dickstein et al., 2015), SMG (Song & Ermon, 2019; 2020), and SDE (Song et al., 2021) have been widely adapted to various domains (Xie et al., 2023; Mao et al., 2025). For instance, within the scope of GNNs, researchers explore to employ DMs to enhance molecular graph modeling (Huang et al., 2023b;a), protein design (Zhang et al., 2023b; Gruver et al., 2023), drug discovery (Guan et al., 2023; Schneuing et al., 2022), material design (Xie et al., 2022), etc, and achieve significant improvement. There also has been an exploration in GAD, Diga (Li et al., 2023) proposes a semi-supervised DM to detect money laundering, where the DM is guided by labeled anomalies for subgraph recovery. GODM (Liu et al., 2023) introduces a plug-and-play package to adopt a variational encoder and diffusion model to generate effective negative samples to solve the class imbalance.

Compared with GODM (Liu et al., 2023), which aims to utilize the powerful generative ability of DMs to generate sufficient negative samples to enhance the performance of current GAD methods, we directly employ DMs to detect the abnormal targets as an end-to-end model with careful designation.

## B    VISUALIZATION

In this section, we visualize the distribution of the reconstructed representations by using t-SNE (Van der Maaten & Hinton, 2008). For a fair demonstration, we select the smaller-scale Books and the larger-scale Weibo dataset and illustrate the results in Figure 6 and Figure 7, respectively. Moreover, for clearer demonstration, we refine the visualization of Books in Figure 8.

Specifically, we visualize the t-SNE results of the reconstructed general, common, and discriminative content with small timestep (t=100) and large timestep (t=400). Additionally, we compare these outcomes against the learned representations derived from the SOTA reconstruction-based method, DOMINANT, and can draw several observations:

(**1**) Current reconstruction-based methods tend to produce highly condensed clusters of learned representations, thereby constraining the discriminative information inherent between distinct nodes. While DM significantly enhances the ability to capture the distribution patterns of samples.
(**2**) Compared to the general content, the common content emphasizes the shared attributes across disparate samples, leading to a clustering effect. Conversely, discriminative content mines the differences, making a more discernible distribution of samples.
(**3**) Discriminative Content exhibits superior performance in identifying anomalous samples, with them distributed at the boundary of the data distributions. This contrasts with the more even distribution of other content categories.

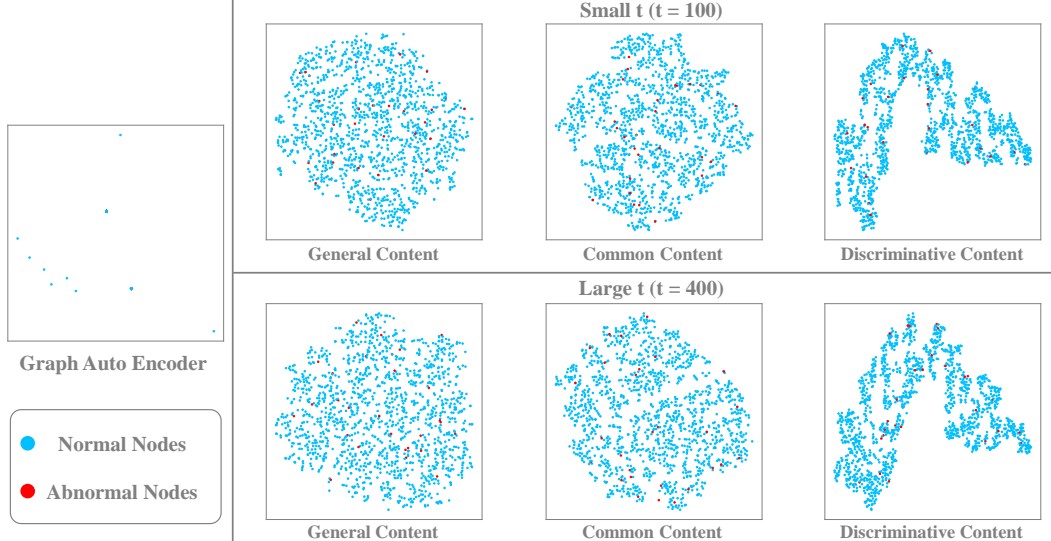

Figure 6: Visualization results of the learned representations by (1) reconstruct-based method (*e.g.* *Dominant*), and (2) different components of DiffGAD on the Books dataset. Variances of points in Graph Auto Encoder: [6.066866e-22, 3.768135e-35], very close to 0.

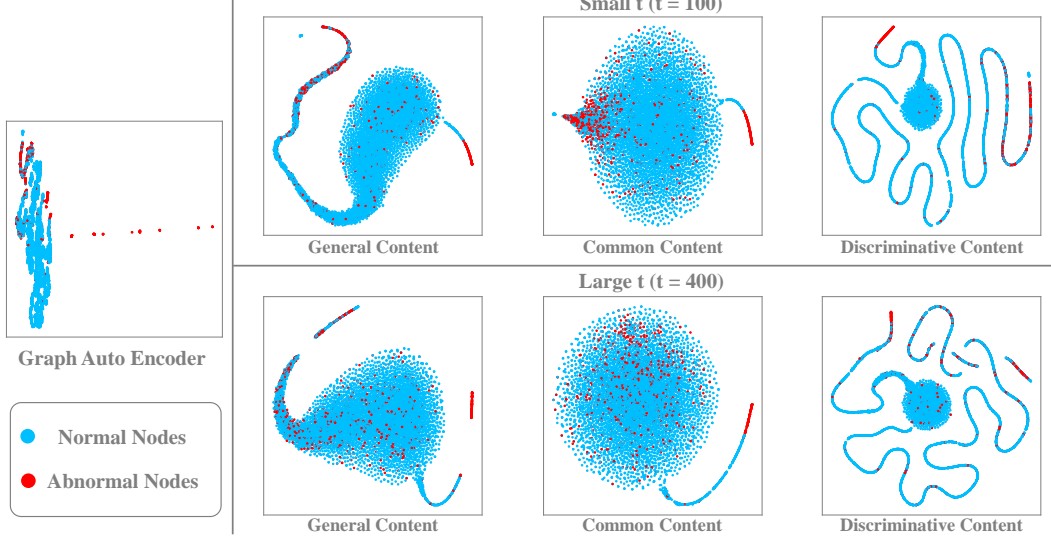

Figure 7: Visualization results of the latent space constructed by (1) reconstruct-based method (*e.g.* *Dominant*), and (2) different components of DiffGAD on the Weibo dataset.

## C  MORE EXPERIMENTAL DETAILS

### C.1  DETAILED EXPERIMENTAL ANALYSIS

In this subsection, we analyze the experimental results of our DiffGAD over the datasets where we achieve sub-optimal performance.

**Weibo.** The success of most methods on Weibo is because the outliers in Weibo exhibit the properties of both structural and contextual outliers. Specifically, in Weibo, the average clustering coefficient of the outliers is higher than that of inliers (0.400 vs. 0.301), meaning that these outliers correspond to structural outliers. Meanwhile, the average neighbor feature similarity of the outliers is far lower than that of inliers (0.004 vs. 0.993), so the outliers also correspond to contextual outliers.

| General Content | Common Content | Discriminative Content |
|---|---|---|

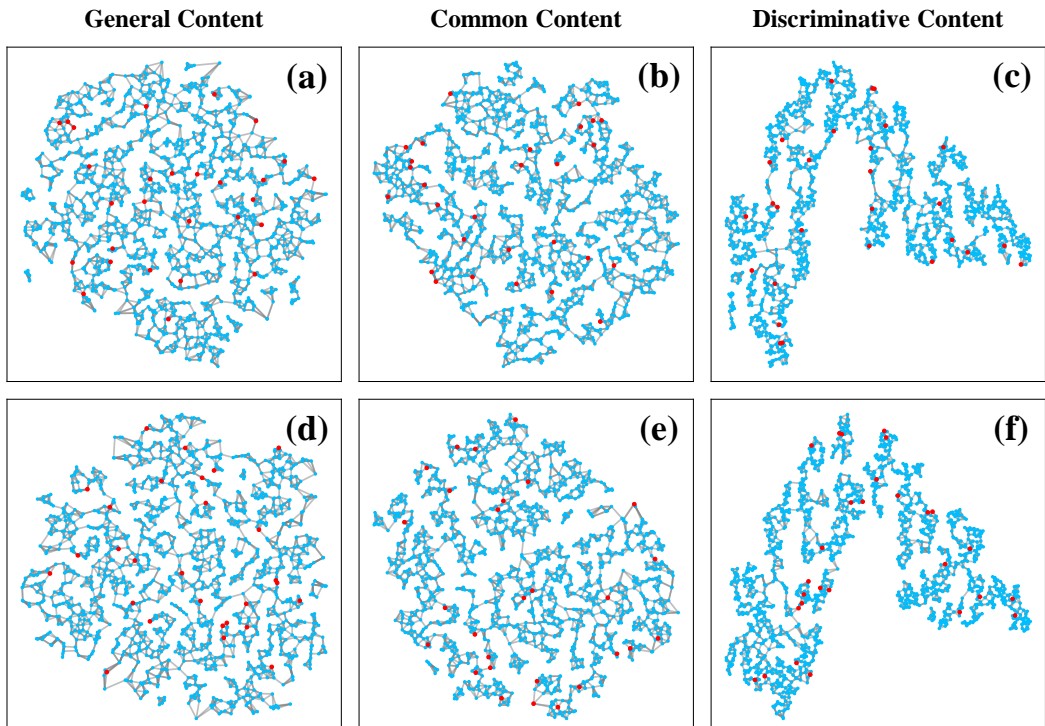

Figure 8: Visualization results of the learned representations by different components of DiffGAD on the Books dataset, for clear illustration, we construct the boundary by connecting neighbors and highlight the abnormal samples. Where (a), (b), (c) are with small t (t=100), and (d), (e), (f) are with large t (t=400), please zoom in for clearer observation.

The two classical algorithms Radar and ANOMALOUS employ graph-based features to extract the anomalous signals from graphs to more flexibly encode graph information to spot outlier nodes, which helps them perform best on Weibo. As they constraint on graph/node types or prior knowledge, they can not generalize well on other datasets.

**Reddit & Dgraph.** In contrast, the outliers in the Reddit and DGraph datasets have similar average neighbor feature similarities and clustering coefficients for outliers and inliers. Therefore, their abnormalities rely more on outlier annotations with domain knowledge.

Non-graph algorithm LOF (Local Outlier Factor) identifies local outliers based on density while IF (Isolation Forest) builds an ensemble of base trees to isolate the data points and defines the decision boundary as the closeness of an individual instance to the root of the tree. Both of them solely use node attributes thus avoiding the influence of structures.

**Disney.** DiffGAD and most of the deep algorithms do not work particularly well on Disney compared to classical baselines. The reason is that Disney has small graphs in terms of 'Nodes', 'Edges', and 'Features' (See Table 5 ). The small amount of data could make it difficult for the deep learning methods to encode the inlier distribution well and could also possibly lead to overfitting issues.

### C.2 DETAILED DATASET DESCRIPTIONS

The detailed statistics of the datasets are shown in Table 5.

**Weibo (Zhao et al., 2020):** Weibo describes the relationship between users, posts, and hashtags on the social platform Tencen-Weibo, where the hashtags serve as the edges, the post's information and the text's bag-of-word vectors make up for the node features. Temporal information is used to label data and the users are suspicious if they post as frequently as bots.

Table 5: The statistics of datasets.

| Dataset | #Nodes | #Edges | #Features | Avg.Degree | Ratio |
|---------|--------|--------|-----------|------------|-------|
| **Weibo** | 8405 | 407963 | 400 | 48.5 | 10.3% |
| **Reddit** | 10984 | 168016 | 64 | 15.3 | 3.3% |
| **Disney** | 124 | 335 | 28 | 2.7 | 4.8% |
| **Books** | 1418 | 3695 | 21 | 2.6 | 2.0% |
| **Enron** | 13533 | 176987 | 18 | 13.1 | 0.4% |
| **Dgraph** | 3700550 | 4300999 | 17 | 1.2 | 0.4% |

**Reddit (Kumar et al., 2019; Wang et al., 2021):** Reddit consists of user posts on subreddits within one month, and the text of each post is transferred to a feature vector to represent the LIWC categories (Pennebaker et al., 2001). The features of users and subreddits are derived by summing the features of respective posts, where the banned users on the platform are considered to be anomalous.

**Disney and Books (Sánchez et al., 2013):** Disney and Books are co-purchase networks of movies and books from Amazon co-purchase networks respectively (Leskovec et al., 2007). Whose node features both include prices, ratings, number of reviews, etc. The ground truth labels of Disney are manually annotated according to the majority vote of high school students and those of Books are derived from amazonfail tag information.

**Enron (Sánchez et al., 2013):** Enron denotes the email network extracted from (Klimt & Yang, 2004), where the email addresses having spam messages are regarded as anomalies. The nodes are composed of emails, whose features describe the average number of recipients, the time range between two emails, and so on.

**Dgraph (Huang et al., 2022):** Dgraph is a large-scale attributed graph with 3M nodes, 4M dynamic edges, and 1M ground-truth nodes. A node represents a financial user, and an edge from one user to another means that the user regards the other user as the emergency contact person. Users who exhibit at least one fraud activity, such as not repaying the loans a long time after the due date and ignoring the platform's repeated reminders, are defined as anomalies/fraudsters.

## C.3    MORE EXPERIMENTAL COMPARISONS

In this subsection, we show more experimental metrics, including Average Precision (AP), Recall@k, and Area Under the Precision and Recall Curve (AUPRC), and the experimental results are listed in Table 6, 7, and 8, respectively. Specifically, we can observe that our DiffGAD achieves favorable performances with small $std$ results around all benchmarks, which demonstrates our effectiveness.

## C.4    MORE ABLATION RESULTS

In this subsection, we provide more ablation results to validate our method. Specifically, we empirically explore the validity of the generated common feature $c_{next}$, and show the distance between $c_{next}$ with the reconstructed Abnormal and Normal features statistically, the L2 distances are shown in Table 9, and the Standardized cosine distances are depicted in Table 10. We can observe that $c_{next}$ is more similar to the normal samples from both results, which shows that normal samples contain more common attributes.

Furthermore, we show more experimental results with different $\lambda$ in Figure 9. We can observe that both the Reddit and Weibo datasets are not sensitive with different $\lambda$ ($\lambda < 2$), but the performance of both datasets drops with $\lambda = 2$, borrowing from the classifier-free guidance (Ho & Salimans, 2022), we believe that $\lambda$ serves as a trading factor, where Reddit needs discriminative content from $\lambda = 0.8$, and $\lambda = 1.0$ works better for Weibo.

Moreover, we also supplement more experimental results with different timestep $t$ in Figure 10. We can observe that Reddit is not sensitive to different $t$ (values from the $y$ axis), and small-scale general content works better for the Enron dataset.

Table 6: Performance comparison (Average Precision, **AP**) among 13 algorithms on 6 datasets, where we show the *avg perf.* $\pm$ *the std of perf.* of each method. The best results of all methods are indicated in boldface, and the second best results are underlined. OOM refers to out-of-memory with CPU. TLE denotes time limit of 24 hours exceeded.

| Algorithm | Weibo | Reddit | Disney | Books | Enron | Dgraph |
|---|---|---|---|---|---|---|
| *graph-agnostic* | | | | | | |
| LOF | $15.8 \pm 0.0$ | $\mathbf{4.2 \pm 0.0}$ | $5.2 \pm 0.0$ | $1.5 \pm 0.0$ | $0.0 \pm 0.0$ | TLE |
| IF | $12.9 \pm 2.6$ | $2.8 \pm 0.1$ | $\mathbf{10.1 \pm 4.5}$ | $1.9 \pm 0.2$ | $0.1 \pm 0.0$ | $\underline{1.8 \pm 0.0}$ |
| MLPAE | $52.8 \pm 9.9$ | $3.4 \pm 0.0$ | $5.9 \pm 0.8$ | $1.8 \pm 0.3$ | $0.1 \pm 0.0$ | $0.9 \pm 0.0$ |
| *classical algorithms* | | | | | | |
| SCAN | $17.3 \pm 3.4$ | $3.3 \pm 0.0$ | $5.0 \pm 0.3$ | $2.0 \pm 0.1$ | $0.0 \pm 0.0$ | TLE |
| Radar | $\mathbf{92.1 \pm 0.7}$ | $3.6 \pm 0.2$ | $7.2 \pm 0.0$ | $2.2 \pm 0.0$ | $\underline{0.2 \pm 0.0}$ | OOM |
| ANOMALOUS | $\mathbf{92.1 \pm 0.7}$ | $\underline{4.0 \pm 0.6}$ | $7.2 \pm 0.0$ | $2.2 \pm 0.0$ | $\underline{0.2 \pm 0.0}$ | OOM |
| *deep algorithms* | | | | | | |
| GCNAE | $70.8 \pm 5.0$ | $3.4 \pm 0.0$ | $4.8 \pm 0.7$ | $2.1 \pm 0.4$ | $0.1 \pm 0.0$ | $1.0 \pm 0.0$ |
| DOMINANT | $18.0 \pm 10.2$ | $3.7 \pm 0.0$ | $\underline{7.6 \pm 5.0}$ | $2.2 \pm 0.6$ | $0.1 \pm 0.1$ | OOM |
| DONE | $65.5 \pm 13.4$ | $3.7 \pm 0.4$ | $5.0 \pm 0.7$ | $1.8 \pm 0.3$ | $0.1 \pm 0.0$ | OOM |
| AdONE | $62.9 \pm 9.5$ | $3.3 \pm 0.4$ | $6.1 \pm 1.5$ | $2.5 \pm 0.3$ | $0.1 \pm 0.0$ | OOM |
| AnomalyDAE | $38.5 \pm 22.5$ | $3.7 \pm 0.1$ | $5.7 \pm 0.2$ | $\underline{3.5 \pm 1.4}$ | $0.1 \pm 0.0$ | OOM |
| GAAN | $80.3 \pm 0.2$ | $3.7 \pm 0.1$ | $5.6 \pm 0.0$ | $2.6 \pm 0.8$ | $0.1 \pm 0.0$ | OOM |
| CONAD | $15.6 \pm 6.9$ | $3.7 \pm 0.3$ | $6.0 \pm 1.4$ | $2.5 \pm 0.8$ | $0.1 \pm 0.0$ | $0.9 \pm 0.0$ |
| **DiffGAD** | $\underline{80.9 \pm 1.7}$ | $3.8 \pm 0.0$ | $6.2 \pm 0.0$ | $\mathbf{8.1 \pm 2.8}$ | $\mathbf{1.7 \pm 4.7}$ | $\mathbf{57.5 \pm 0.0}$ |

Table 7: Performance comparison (**Recall@k%**) among 13 algorithms on 6 datasets, where we show the *avg perf.* $\pm$ *the std of perf.* of each method. The best results of all methods are indicated in boldface, and the second best results are underlined. OOM refers to out-of-memory with CPU. TLE denotes time limit of 24 hours exceeded.

| Algorithm | Weibo | Reddit | Disney | Books | Enron | Dgraph |
|---|---|---|---|---|---|---|
| *graph-agnostic* | | | | | | |
| LOF | $22.0 \pm 0.0$ | $\mathbf{4.4 \pm 0.0}$ | $0.0 \pm 0.0$ | $0.0 \pm 0.0$ | $0.0 \pm 0.0$ | TLE |
| IF | $13.8 \pm 6.4$ | $0.1 \pm 0.1$ | $\mathbf{9.2 \pm 8.3}$ | $1.1 \pm 1.6$ | $0.0 \pm 0.0$ | $0.1 \pm 0.1$ |
| MLPAE | $48.9 \pm 11.0$ | $3.0 \pm 0.0$ | $0.0 \pm 0.0$ | $0.9 \pm 1.6$ | $0.0 \pm 0.0$ | $\underline{0.5 \pm 0.1}$ |
| *classical algorithms* | | | | | | |
| SCAN | $23.8 \pm 7.0$ | $2.7 \pm 0.3$ | $\underline{7.5 \pm 11.2}$ | $0.7 \pm 1.4$ | $0.0 \pm 0.0$ | TLE |
| Radar | $\mathbf{86.4 \pm 0.8}$ | $2.1 \pm 0.8$ | $0.0 \pm 0.0$ | $0.0 \pm 0.0$ | $0.0 \pm 0.0$ | OOM |
| ANOMALOUS | $\mathbf{86.4 \pm 0.8}$ | $\underline{4.0 \pm 1.9}$ | $0.0 \pm 0.0$ | $0.0 \pm 0.0$ | $0.0 \pm 0.0$ | OOM |
| *deep algorithms* | | | | | | |
| GCNAE | $67.6 \pm 5.2$ | $3.0 \pm 0.0$ | $0.0 \pm 0.0$ | $0.7 \pm 1.8$ | $0.0 \pm 0.0$ | $0.4 \pm 0.0$ |
| DOMINANT | $19.7 \pm 13.8$ | $0.9 \pm 0.4$ | $3.3 \pm 6.7$ | $1.6 \pm 3.1$ | $0.0 \pm 0.0$ | OOM |
| DONE | $65.4 \pm 12.4$ | $2.8 \pm 1.6$ | $0.0 \pm 0.0$ | $1.1 \pm 1.6$ | $0.0 \pm 0.0$ | OOM |
| AdONE | $64.3 \pm 7.6$ | $1.0 \pm 1.2$ | $1.7 \pm 5.0$ | $3.0 \pm 1.7$ | $0.0 \pm 0.0$ | OOM |
| AnomalyDAE | $42.2 \pm 23.7$ | $0.9 \pm 0.5$ | $0.0 \pm 0.0$ | $\underline{2.7 \pm 2.2}$ | $0.0 \pm 0.0$ | OOM |
| GAAN | $77.1 \pm 0.2$ | $1.1 \pm 0.4$ | $0.0 \pm 0.0$ | $1.8 \pm 1.8$ | $0.0 \pm 0.0$ | OOM |
| CONAD | $20.3 \pm 13.3$ | $1.3 \pm 1.6$ | $0.8 \pm 3.6$ | $1.7 \pm 2.9$ | $0.0 \pm 0.0$ | $0.4 \pm 0.1$ |
| **DiffGAD** | $\underline{77.1 \pm 0.5}$ | $2.3 \pm 1.2$ | $0.0 \pm 0.0$ | $\mathbf{13.8 \pm 2.9}$ | $\mathbf{3.0 \pm 7.3}$ | $\mathbf{57.3 \pm 0.0}$ |

## C.5 ADAPTATION WITH DIFFERENT BACKBONES.

In this subsection, we evaluate the adaptation ability of DiffGAD with different backbones, Specifically, we utilize 4 different encoder & decoder architectures, including VGAE, MLP, VAE, Graph Transformer, and evaluate the ROC-AUC result. The experimental results on the backbones are listed in Table 11, and results on DiffGAD with different backbones are shown in Table 12. From the experimental results, we can draw following observations:

Table 8: Performance comparison (Area under Precision Recall Curve, **AUPRC**) among deep learning based algorithms on 5 datasets, where we show the *avg perf.* ± *the std of perf.* of each method. The best results of all methods are indicated in boldface, and the second best results are underlined.

| Algorithm | Weibo | Reddit | Disney | Books | Enron |
|---|---|---|---|---|---|
| GCNAE | $76.5 \pm 6.9$ | $3.4 \pm 0.0$ | $4.0 \pm 0.5$ | $1.9 \pm 0.2$ | $0.0 \pm 0.0$ |
| DOMINANT | $55.6 \pm 9.4$ | $3.7 \pm 0.0$ | $\mathbf{6.0 \pm 0.7}$ | $1.6 \pm 0.1$ | $0.1 \pm 0.0$ |
| DONE | $64.9 \pm 1.9$ | $3.7 \pm 0.0$ | $3.7 \pm 0.0$ | $1.5 \pm 0.0$ | $0.0 \pm 0.0$ |
| AdONE | $67.7 \pm 0.9$ | $3.4 \pm 0.0$ | $4.2 \pm 0.0$ | $2.5 \pm 0.0$ | $0.1 \pm 0.0$ |
| AnomalyDAE | $72.2 \pm 10.8$ | $3.7 \pm 0.0$ | $4.8 \pm 0.9$ | $\mathbf{11.6 \pm 17.2}$ | $0.1 \pm 0.0$ |
| CONAD | $61.0 \pm 27.4$ | $3.7 \pm 0.0$ | $5.4 \pm 3.7$ | $2.2 \pm 0.5$ | $0.1 \pm 0.0$ |
| **DiffGAD** | $\mathbf{80.9 \pm 1.7}$ | $\mathbf{3.7 \pm 0.0}$ | $\underline{5.0 \pm 0.0}$ | $7.8 \pm 2.3$ | $\mathbf{1.4 \pm 4.5}$ |

Table 9: L2 distance (↓) of $c_{next}$ to the reconstructed abnormal and normal samples, respectively. The distance is calculated as the mean of all timesteps.

| L2 Dis (↓) | Weibo | Reddit | Disney | Books | Enron |
|---|---|---|---|---|---|
| Abnormal | 295.16 | 0.4223 | 2611.96 | 0.9888 | 0.0718 |
| Normal | **18.05** | **0.4209** | **2372.6** | **0.9886** | **0.0704** |

1. GAE and DiffGAD(GAE) consistently exhibit superior performance compared to different encoder & decoder architectures across different datasets in terms of ROC-AUC. Such results underscore the robustness of GAE and the generalization ability of DiffGAD.

2. Without encoding graph structure, the DiffGAD(MLP) achieves notable results on the Enron dataset, this indicates that the graph structure can lead to a loss of discriminative information, as MLPs rely solely on node features as input.

3. The different performance of GAE, VAE, and VGAE reflects the impact of different encoder and decoder architectures. However, the architecture of Graph Transformer may be somewhat excessive for this task, as it exhibits a relatively high complexity, but without demonstrating any performance improvement.

To sum up: we find that (1) DiffGAD achieves robust performance gains over different architectures, which demonstrates its adaptation ability, and (2) a simple architecture (i.e. GAE) is enough for latent space construction over current datasets, which preserves sufficient discriminative information, and further works with this latent space can boost the detection performance.

## C.6   CHOICES ABOUT $\lambda$.

In this subsection, we describe the selection of the key hyper-parameter $\lambda$. Specifically, we statistic the heterophily of the Abnormal samples over different dataset, and the results are listed in Table 13. We observe that on the dataset with larger anomaly heterophily, such as Enron, Disney, and Books, larger $\lambda$ works well, while on the dataset with smaller anomaly heterophily, like Weibo and Reddit, smaller $\lambda$ works well. To summarize, we draw the following conclusion:

1. Larger $\lambda$ works better on the datasets where anomaly samples has larger heterophily, and we recommend using $\lambda$ larger than 1 on such data. Larger heterophily for anomaly samples means they are hidden within normal samples, and thus they are harder to detect.

2. Smaller $\lambda$ works well on the datasets where anomaly samples has smaller heterophily, and we recommend using $\lambda$ smaller than 1 on such data. Smaller heterophily for anomaly samples means they are clustered together, and thus they are easier to detect.

Table 10: Standardized Cosine distance ($\uparrow$) of $c_{next}$ to the reconstructed abnormal and normal samples, respectively. The distance is calculated as the mean of all timesteps.

| Cos Dis ($\uparrow$) | Weibo | Reddit | Disney | Books | Enron |
|---|---|---|---|---|---|
| Abnormal | 0.1828 | 0.4947 | 0.3424 | 0.5006 | 0.4987 |
| Normal | **0.4347** | **0.4949** | **0.4214** | **0.5008** | **0.4998** |

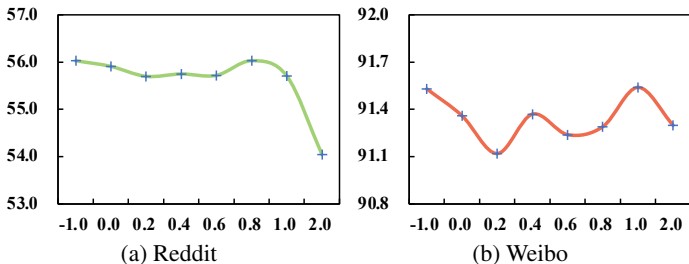

(a) Reddit        (b) Weibo

Figure 9: The ROC-AUC performance of another 2 datasets under the different scale of the control of $\lambda$, where $x$ axis represents the variance of $\lambda$, and $y$ axis is the ROC-AUC results.

## C.7 DETAILED BASELINE DESCRIPTION

**LOF (Breunig et al., 2000):** Local Outlier Factor (LOF) calculates the extent to which the node is anomalous according to how isolated it is compared to its neighborhood. It is worth noting that LOF only uses node features, and the neighborhood is chosen by k-nearest-neighbors (KNN).

**IF (Liu et al., 2012):** Isolation Forest (IF) is a well-established tree ensemble technique employed in anomaly detection, where it constructs an ensemble of base trees to isolate data points. The decision boundary is established based on the proximity of individual instances to the root of each tree. IF only utilizes the node features of the data.

**MLPAE (Sakurada & Yairi, 2014):** MLPAE utilizes the Multilayer Perceptron (MLP) as both the encoder and decoder to reconstruct node features. Specifically, the encoder processes the node features to learn their low-dimensional embedding, and the decoder reconstructs the input node features from these embedding, employing the reconstruction loss of the node features as the anomaly score for each node.

**SCAN (Xu et al., 2007)** The Structural Clustering Algorithm for Networks (SCAN) is a method designed to detect clusters, hub nodes, and outlier nodes in a graph using only the graph's structural information. SCAN identifies structural anomalies by detecting clusters and considering the nodes within these clusters as potential outliers. Since structural outliers exhibit distinct clustering patterns, SCAN is particularly effective for this purpose.

**Radar (Li et al., 2017):** Radar is an anomaly detection framework for attributed graphs that utilizes both graph structure and node features as inputs. It identifies outlier nodes whose behaviors deviate significantly from the majority in terms of the residuals of feature information and coherence with network structure. The anomaly score is determined by the norm of reconstruction residuals.

**ANOMALOUS (Peng et al., 2018):** ANOMALOUS performs both anomaly detection and attribute selection on attributed graphs using CUR decomposition and residual analysis. Similar to Radar, the anomaly score is also determined by the norm of its reconstruction residuals.

**GCNAE (Kipf & Welling, 2016):** GCNAE is an AE framework that employs GCN as both the encoder and decoder. The encoder takes graph structure and node features as inputs to aggregate information from a node's neighbors to learn its latent representation and the decoder uses the GCN to reconstruct the node features and graph structure from the embedding. The anomaly score for each node is determined by the reconstruction error of the decoder.

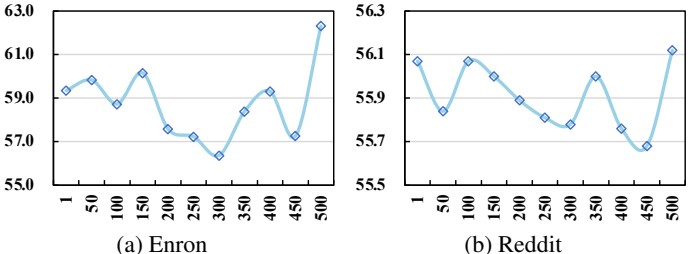

(a) Enron                                    (b) Reddit

Figure 10: The ROC-AUC performance of different timesteps $t$ over another 2 representative datasets, where $x$ axis represents different timesteps $t$, and $y$ axis is the ROC-AUC results.

Table 11: ROC-AUC result of different encoder-encoder architectures on 5 datasets, where we show the *avg perf.* ± *the std of perf.* of each method. The best results of all methods are indicated in boldface.

| Backbone | Weibo | Reddit | Disney | Books | Enron | Avg. |
|---|---|---|---|---|---|---|
| MLP | $85.5 \pm 0.3$ | $50.6 \pm 0.0$ | $\mathbf{48.1 \pm 0.1}$ | $49.3 \pm 5.9$ | $41.3 \pm 14.1$ | $55.0 \pm 4.1$ |
| VAE | $84.4 \pm 0.2$ | $50.6 \pm 0.0$ | $\mathbf{48.0 \pm 0.1}$ | $50.5 \pm 5.4$ | $42.9 \pm 8.2$ | $55.3 \pm 2.8$ |
| VGAE | $\mathbf{92.5 \pm 0.0}$ | $55.8 \pm 0.1$ | $47.8 \pm 3.3$ | $48.4 \pm 4.2$ | $56.8 \pm 0.7$ | $60.3 \pm 1.7$ |
| GTrans | $90.9 \pm 0.4$ | $\mathbf{56.1 \pm 0.0}$ | $31.0 \pm 7.0$ | $32.5 \pm 12.3$ | $55.5 \pm 0.6$ | $53.2 \pm 4.1$ |
| GAE | $92.0 \pm 0.1$ | $\mathbf{56.1 \pm 0.0}$ | $40.3 \pm 6.7$ | $\mathbf{59.1 \pm 2.5}$ | $\mathbf{59.1 \pm 1.6}$ | $\mathbf{61.3 \pm 2.2}$ |

**DOMINANT (Ding et al., 2019):** DOMINANT is a pioneering work that leverages GCN and AE for anomaly detection. It follows an encoder-decoder architecture with a two-layer GCN. The node feature decoder is also a two-layer GCN, while the graph structure is decoded by a one-layer GCN and dot product. The anomaly score of each node is determined by the weighted sum of two decoders.

**DONE (Bandyopadhyay et al., 2020):** DONE leverages an MLP-based encoder-decoder architecture to reconstruct both the adjacency matrix and node features. It employs separate AEs for structural and feature information. The optimization of node embeddings and anomaly scores is performed simultaneously using a unified loss function.

**AdONE (Bandyopadhyay et al., 2020):** AdONE is an extension of DONE, incorporating an additional discriminator to differentiate between the learned structural and feature embedding of a node. This adversarial training strategy aims to achieve better alignment of two distinct embeddings within the latent space.

**AnomalyDAE (Fan et al., 2020):** AnomalyDAE employs both a structural AE and an attribute AE to detect outlier nodes. The encoder encodes adjacency matrix and node features respectively to obtain two embeddings, while the attribute decoder reconstructs the node features using both structural and attribute embeddings.

**GAAN (Chen et al., 2020):** GAAN is a GAN-based method for outlier node detection utilizing an MLP-based generator to create fake graphs and an MLP-based encoder to encode graph information. A discriminator is then trained to distinguish between real and fake graphs. The anomaly score is determined by combining the node reconstruction error and the confidence in identifying real nodes.

**CONAD (Xu et al., 2022):** CONAD leverages graph augmentation and cognitive learning techniques to detect outlier nodes. It generates augmented graphs to impose prior knowledge of anomalies. The graph encoder is optimized through a contrastive learning loss. Similar to DOMINANT, the outlier score is determined by the weighted sum of two different decoders.

## D IMPLEMENTATION DETAILS

**Environment.** The key libraries and their versions used in the experiment are as follows: Python=3.11, CUDA_version=11.8, torch=2.0.1, pytorch_geometric=2.4.0, pygod=0.4.0, numpy=1.25.0

Table 12: ROC-AUC result of DiffGAD with different encoder-encoder architectures on 5 datasets, where we show the *avg perf.* $\pm$ *the std of perf.* of each method. The best results of all methods are indicated in boldface.

| Method | Weibo | Reddit | Disney | Books | Enron | Avg. |
|---|---|---|---|---|---|---|
| DiffGAD(MLP) | $90.0 \pm 1.1$ | $50.6 \pm 0.0$ | $49.1 \pm 0.2$ | $50.0 \pm 0.0$ | $\mathbf{81.9 \pm 5.2}$ | $64.3 \pm 1.3$ |
| DiffGAD(VAE) | $90.5 \pm 0.6$ | $50.6 \pm 0.0$ | $48.5 \pm 0.2$ | $\mathbf{67.1 \pm 2.9}$ | $66.0 \pm 1.8$ | $64.5 \pm 1.1$ |
| DiffGAD(VGAE) | $92.5 \pm 0.0$ | $56.1 \pm 0.0$ | $52.1 \pm 3.2$ | $51.0 \pm 1.1$ | $57.8 \pm 0.2$ | $61.9 \pm 0.5$ |
| DiffGAD(GTrans) | $93.0 \pm 0.7$ | $56.2 \pm 0.1$ | $51.4 \pm 0.6$ | $51.5 \pm 0.5$ | $65.7 \pm 4.9$ | $63.6 \pm 1.4$ |
| DiffGAD(GAE) | $\mathbf{93.4 \pm 0.3}$ | $\mathbf{56.3 \pm 0.1}$ | $\mathbf{54.5 \pm 0.2}$ | $66.4 \pm 1.8$ | $71.6 \pm 7.0$ | $\mathbf{68.4 \pm 1.9}$ |

Table 13: The homophily and heterophily of anomalies on 5 datasets.

| | Weibo | Reddit | Disney | Books | Enron |
|---|---|---|---|---|---|
| **Homophily** | 0.87 | 0.18 | 0.01 | 0.00 | 1.00 |
| **Heterophily** | 0.13 | 0.82 | **0.99** | **1.00** | **0.00** |

**Hardware configuration.** All the experiments were performed on a Linux server with a 3.00GHz Intel Xeon Gold 6248R CPU,1T RAM, and 1 NVIDIA A40 GPU with 45GB memory.

**Model Architectures.** The architectural details of DiffGAD are listed here for easier reproduction. The Graph AE, which is optimized by Adam, uses a two-layer GCN as the encoder and a two-layer GCN decoder to reconstruct the node attribute. The structural decoder utilizes a one-layer GCN and dot product to reconstruct the graph adjacency matrix. As for the DM, the denoising function is a multi-layer MLP with SiLU activations. More details can be found in our code.

**Hyper-parameters.** The hyper-parameters are listed in Table 14, our unconditional DM and conditional DM share the same parameters.

Table 14: Hyper-parameter for different datasets.

| Hyper-paramter | Type | Weibo | Reddit | Disney | Books | Enron | Dgraph |
|---|---|---|---|---|---|---|---|
| Batch size | AE | | | FULL BATCH | | | 8192 |
| Epochs | AE | | | 300 | | | |
| Early stop | AE | | | No | | | |
| Dropout | AE | 0.3 | 0.3 | 0.3 | 0.1 | 0.1 | 0.3 |
| Learning Rate | AE | 0.01 | 0.05 | 0.01 | 0.1 | 0.01 | 0.1 |
| $\alpha$ | AE | 0.8 | 0.8 | 1.0 | 0.5 | 0.0 | 1.0 |
| dimension | AE | 128 | 32 | 8 | 8 | 8 | 8 |
| Batch size | DM | | | FULL BATCH | | | 8192 |
| Epochs | DM | | | 800 | | | |
| Early stop | DM | | | Yes | | | |
| Learning Rate | DM | | | 0.005 | | | |
| dimension | DM | 256 | 64 | 16 | 16 | 16 | 16 |
| $\lambda$ | DM | 1.0 | 0.8 | 2.0 | 2.0 | 2.0 | 1.0 |

