# OpenReview forum: "DiffGAD: A Diffusion-based Unsupervised Graph Anomaly Detector"
_ICLR.cc/2025/Conference — ICLR 2025 Poster_

### Official Review · Reviewer_epRR · 2024-11-01

**Soundness:** 3
**Presentation:** 2
**Contribution:** 2
**Rating:** 5
**Confidence:** 5

**Summary:**

The paper presents a graph anomaly detection method, DiffGAD, which aims to capture critical discriminative content in reconstruction. By leveraging diffusion sampling, DiffGAD infuses the latent space with discriminative content. To evaluate its effectiveness, the authors conduct experiments on seven datasets and compare its performance against other anomaly detection methods.

**Strengths:**

1. This paper presents a clear motivation.
2. The logical structure of the paper is clear.

**Weaknesses:**

1. The paper states that ‘the latent space constructed by the AE-based method (Ding et al., 2019) tends to represent all samples for the Books dataset (Sánchez et al., 2013) into the same point’ and 'VAE constructs the latent space within a constrained distribution (e.g., the Gaussian distribution), leading to a uniform latent distribution.' This argument should be further supported through experimental validation.
2. Authors should add the latest baselines to demonstrate the effectiveness of DiffGAD.
3. Sec. 4.3 should further analyze why DIFFGAD does not achieve optimal performance.

**Questions:**

1. The GAE model detects anomalies by calculating reconstruction error. Wouldn’t it be more effective to focus on reconstructing common attributes rather than distinguishable features, as anomalies are typically identified by their deviations from common patterns?
2. Why add noise to the features? Can’t the original features capture the general patterns?
3. In the visualization results, normal nodes and abnormal nodes are not clearly distinguished.

---

> ### Author Response · Authors · 2024-11-24
> **Response to Reviewer epRR ---- Part 1/3**
>
> **Response to Reviewer $\color{orange}{\text{epRR}}$**
>
> We highly appreciate your insightful comments and thoughtful feedback! Your constructive criticism is invaluable in refining our work.  We notice that your comments and questions are mostly focused on our **experiments** , for clear demonstration, we provide the point-to-point clarification and explanation below. If you have additional questions, we would be pleased to discuss them with you.
>
> > **Comment 1: Experimental validation with Books and VAE methods.** - "... This argument should be further supported through experimental validation."
>
> Thank you for raising this point!  We fully understand your concerns.
>
> **The Result of AE on Books**
>
> Specifically, **we have included the visualization results in Figure 6 (Graph Auto Encoder) of the paper**. For further clarification, we provide the extracted embeddings by AE in Books, and find that, **all of the nodes have the same embedding**:
>
> `"[2.103607, 2.091973, -2.173611, -2.142506, -2.1222112, 2.1332016, 2.14255, -2.141333]" `
>
> Furthermore, we also notice that the t-SNE results of GAE got 10 different points over a total of 1418 samples, to answer this, we utilize PCA analysis, and got the variances of these points as `"[6.066866e-22, 3.768135e-35]"`, the variances of these points are very close to 0, **these difference points can be attributed to the floating-point representation,** we will highlight this in the paper.
>
> **VAE constructs constrained distribution**
>
> **This statement can be referred to Section 3 of [1]** , where VAE constrains the latent space with the equation $z = \mu + \epsilon * exp(log(\sigma)), \epsilon \in \mathcal{N}(0, 1)$, which is parameterized with $\mu$ and  $log(\sigma)$. And, we can observe that the learned latent representation $z$ is modeled as a random variable from a Gaussian distribution with mean $\mu$ and std $exp(log(\sigma))$.
>
> > **Comment 2: Comparison with latest baselines** - "Authors should add the latest baselines..."
>
> Thanks for your invaluable comments! Following your suggestion, we have **conducted additional experiments** on GAD-EBM [2], GDSS_Rec, and GDSS_Energy [3] on 5 datasets. We will add the latest baselines in our revised submission!
>
> **Table 1: ROC-AUC (%) comparison with latest baselines on 5 datasets, where we show the *avg perf.* ± *the std of perf*. TLE indicates that the method exceeded the time limit of 24 hours.**
>
> |             |    Weibo     |    Reddit    |    Disney     |    Books     |    Enron     |     Avg      |    Dgraph    |
> | :---------: | :----------: | :----------: | :-----------: | :----------: | :----------: | :----------: | :----------: |
> |   GAD-EBM   |   84.5±8.5   |   53.8±5.7   |   57.4±15.1   |   62.9±0.7   | **79.7±2.4** |   67.7±6.5   | **60.3±2.5** |
> |  GDSS_Rec   |  74.5±12.6   |   44.5±0.4   | **65.0±11.2** |   57.1±2.8   |   44.0±4.0   |   57.0±6.2   |     TLE      |
> | GDSS_Energy |  51.9±11.1   |   55.1±0.8   |   58.4±5.9    |   52.7±3.0   |   36.5±5.6   |   50.9±5.3   |     TLE      |
> |   DiffGAD   | **93.4±0.3** | **56.3±0.1** |   54.5±0.2    | **66.4±1.8** |   71.6±7.0   | **68.4±1.9** |   52.4±0.0   |
>
> We can observe that:
>
> - DiffGAD consistently exhibits superior performance compared to an energy-based method and score-based baseline across varying datasets in terms of ROC-AUC metric. Such results underscore the robustness of DiffGAD.
> - GAD-EBM slightly edges out DiffGAD on Enron and Dgraph, a potential explanation lies in that GAD-EBM takes ego-subgraph presentation into consideration with an energy-based model, which explicitly utilizes more structural information.
>
> - GDSS_Rec and GDSS_Energy directly apply the score-based diffusion model to graph features and structures to calculate the reconstruction errors, which show the best result on small dataset Disney but poor results on larger ones. We attribute this to the direct utilization of node features and structural information by the diffusion model.

---

> ### Author Response · Authors · 2024-11-24
> **Response to Reviewer epRR ---- Part 2/3**
>
> > **Comment 3: Experimental analysis about sub-optimal performance** - "...further analyze why DIFFGAD does not achieve optimal performance."
>
> Thanks for your guidance! In light of this, we further analyze the performance below and will add in our revised version.
>
> - **Weibo**
>
> The success of most methods on Weibo is because the outliers in Weibo exhibit the properties of both structural and contextual outliers. Specifically, in Weibo, the average clustering coefficient of the outliers is higher than that of inliers **(0.400 vs. 0.301)**, meaning that these outliers correspond to structural outliers. Meanwhile, the average neighbor feature similarity of the outliers is far lower than that of inliers **(0.004 vs. 0.993)**, so the outliers also correspond to contextual outliers.
>
> The two classical algorithms Radar and ANOMALOUS employ graph-based features to extract the anomalous signals from graphs to more flexibly encode graph information to spot outlier nodes, which helps them perform best on Weibo. As they are constraint on graph/node types or prior knowledge, they can not generalize well on other datasets.
>
> - **Reddit & Dgraph**
>
> In contrast, the outliers in the Reddit and DGraph datasets have similar average neighbor feature similarities and clustering coefficients for outliers and inliers. Therefore, their abnormalities rely more on outlier annotations with domain knowledge.
>
> Non-graph algorithm LOF (Local Outlier Factor) identifies local outliers based on density while IF (Isolation Forest) builds an ensemble of base trees to isolate the data points and defines the decision boundary as the closeness of an individual instance to the root of the tree. Both of them solely use node attributes thus avoiding the influence of structures.
>
> - **Disney**
>
> DiffGAD and most of the deep algorithms do not work particularly well on Disney compared to classical baselines. The reason is that Disney has small graphs in terms of 'Nodes', 'Edges', and 'Features' (See Table 5 in our paper). The small amount of data could make it difficult for the deep learning methods to encode the inlier distribution well and could also possibly lead to overfitting issues.
>
>
>
> > **Question 1: Illustration about reconstructing common and discriminative attributes.** - " Wouldn’t it be more effective to focus on reconstructing common attributes rather than distinguishable features?"
>
> Thank you for highlighting this! Specifically, in this work, common content denotes the shared content across all node features, and discriminative content refers to the distinguishable attributes across different features. Through the connectivity of edges, the common content between the abnormal and normal samples is spread. And thus making the common attributes dominant the reconstruction process. Though reconstructing a large amount of common content makes the model converge, it does destroy the discriminative ability of the model, where both normal and abnormal features draw similar reconstruction scores. For example, in fraud detection, the common content like ages, and interaction information is easy to construct, while the discriminative content like user behavior is difficult to reconstruct, since behavior is diverse across users, but it's the key for detection.
>
>    Moreover, in the **Table 1** of our paper, we also show the experimental results by reconstructing different components, where the 'Diff' shows reconstruction by general content, 'Cond-Diff' shows reconstruction by common content, 'DiffGAD' shows reconstruction by discriminative content, we can draw that:
>
> 1. The similar results of 'Cond-Diff' and 'Diff' prove our hypothesis that common attributes dominate the reconstruction process.
>
> 2. The significant improvements of 'DiffGAD' compared to 'Cond-Diff' and 'Diff' further show the effectiveness of reconstructing the distilled discriminative content.
>
> Meanwhile, from our visualization (Figure 6 & Figure 7 in our paper), we can also draw the same conclusions:
>
> 1. The similar results of 'general' and 'common' demonstrate that the reconstruction process is dominated by common content, and thus it inclines to learn a unified space (common attributes).
> 2. The significant differences between discriminative with general & common show the effectiveness of distilling discriminative content, where we can observe that differences between normal samples are also significant. We can also observe that the abnormal samples are almost located outside the overall distribution, which further illustrates our effectiveness.

---

> ### Author Response · Authors · 2024-11-24
> **Response to Reviewer epRR ---- Part 3/3**
>
> > **Question 2: Illustration of the perturbutation** - "Why add noise to the features?"
>
> **The Role of Perturbutation**
>
> In our work, we add noises to retain general content from different scales, where a lower t preserves more general content. Given that some discriminative content can be challenging to extract, the purpose of *t* is to eliminate certain common elements within the general content, thereby facilitating a clearer distinction of the discriminative features. It is important to note that general content encompasses both common and discriminative elements.
>
> For further demonstration, we conduct experiments utilizing the original feature for discriminative distillation, and the results are listed as follows.
>
> **Table 2: The ROC-AUC (%) performance of VAE architectures with our DiffGAD, where we show the *avg perf.* ± *the std of perf*.**
>
> |    Method    |    Weibo     |    Reddit    |    Disney    |    Books     | Enron        |     Avg      |
> | :----------: | :----------: | :----------: | :----------: | :----------: | ------------ | :----------: |
> |     GAE      |   92.0±0.1   |   56.1±0.0   |   40.3±6.7   |   59.1±2.5   | 59.1±1.6     |   61.3±2.2   |
> |   Original   |   91.8±0.3   |   55.5±1.2   |   51.4±2.4   |   58.3±2.3   | 58.5±2.5     |   63.1±1.7   |
> | DiffGAD(GAE) | **93.4±0.3** | **56.3±0.1** | **54.5±0.2** | **66.4±1.8** | **71.6±7.0** | **68.4±1.9** |
>
> We observe that directly employing the original feature can't improve the performance, the reason can be attributed to the nature of the diffusion model, which aims to denoise from the noisy input, and then reconstruct the original inputs [4,5,6]. Directly denoising from the original feature can cause information corruption, where some key information is removed with the denoising process.
>
>
>
> > **Question 3: Illustration of the visualization** - "In the visualization results, the normal and abnormal nodes are not clearly distinguished."
>
> Thank you for your kind suggestions.  We believe that a large amount of common content between normal and abnormal samples makes it unlikely to distinctly separate them into two clearly defined clusters. This also indicates that this is a very challenging task. If the dataset were easily separable, it would be easy for current methods to draw high detection results. On the Weibo dataset, with a 93% AUC result, we can observe that our method constructs a clear boundary between abnormal and normal samples. On the more challenging books dataset, for convenient illustration, we will construct the boundary by connecting neighbors and highlighting the abnormal samples in the revised pdf. We can clearly observe that our method are able to push the abnormal nodes to the boundary, which indicates our effectiveness.
>
>
>
> [1] Kingma, Diederik P. "Auto-encoding variational bayes." In arXiv 2013.
>
> [2] Amit Roy, et al. GAD-EBM: Graph Anomaly Detection using Energy-Based Models. In NeurIPS 2023.
>
> [3] Dmitrii Gavrilev, et al. Anomaly Detection in Networks via Score-Based Generative Models. In ICML 2023.
>
> [4] Jonathan Ho, et al. Denoising Diffusion Probabilistic Models. In NeurIPS 2020.
>
> [5] Tero Karras, et al. Elucidating the Design Space of Diffusion-Based Generative Models. In NeurIPS 2022.
>
> [6] Yang Song, et al. Score-Based Generative Modeling through Stochastic Differential Equations. In ICLR 2021.

---

> ### Comment · Reviewer_epRR · 2024-11-26
>
> Thank you to the authors for the response, which has largely addressed my concerns.

---

> > ### Author Response · Authors · 2024-11-26
> > **Heartfelt Gratitude to Reviewer epRR！**
> >
> > Dear Reviewer $\color{orange}{\text{epRR}}$,
> >
> > Thanks a lot for your acknowledgment of our efforts! We are thrilled to hear that your concerns have been effectively addressed. We sincerely appreciate your dedicated time and effort in reviewing our updated work and offering invaluable and positive feedback. If you have any further questions, please do not hesitate to contact us. We would be more than happy to continue the discussion with you.
> >
> > Best regards,
> >
> > Authors of Paper 6534

---

> > ### Author Response · Authors · 2024-12-02
> > **Greatly appreciate your support!**
> >
> > Dear Reviewer $\color{orange}{\text{epRR}}$,
> >
> > We are extremely grateful for the insightful comments you provided and for the recognition you have shown toward our work.
> >
> > As the discussion phase comes to a close, **if our response has adequately addressed most of your concerns, we humbly ask if you might consider the possibility of raising your score.** Your support at this stage is immensely important to us!
> >
> > We are fully committed to advancing the field of Graph Anomaly Detection and contributing meaningfully to the community. Your feedback and support are invaluable to us in achieving this goal.
> >
> > Best regards,
> >
> > Authors of Paper 6534

---

### Official Review · Reviewer_2tPm · 2024-11-01

**Soundness:** 3
**Presentation:** 2
**Contribution:** 2
**Rating:** 6
**Confidence:** 3

**Summary:**

This paper proposes DiffGAD, an unsupervised graph anomaly detector based on diffusion models, designed to address the limited capability of capturing discriminative content in graph anomaly detection.

The main contributions include:

Pioneering the application of diffusion models in graph anomaly detection by adapting them from generative tasks, presenting DiffGAD to enhance the model's discriminative ability.
Proposing a generative paradigm guided by discriminative content to extract and refine discriminative features into the latent space, and designing a content preservation strategy to improve the reliability of the guidance process.
Demonstrating the effectiveness of DiffGAD through extensive experiments on 6 real-world datasets and 13 baseline methods.

**Strengths:**

1、The authors innovatively adapt diffusion models from generative tasks to the field of graph anomaly detection, introducing DiffGAD, which brings new perspectives to graph anomaly detection.

2、A novel latent space learning paradigm is introduced by combining unconditional and conditional diffusion models to capture and refine discriminative content, offering a fresh approach to address the challenge of extracting discriminative features in graph anomaly detection.

3、When addressing anomaly detection, the method incorporates diffusion sampling and a content preservation mechanism, effectively injecting and retaining discriminative content across different scales.

4、In the experimental section, six different real-world and large-scale datasets are used for evaluation, covering various types of data scenarios, such as social networks (Weibo, Reddit), commercial networks (Disney, Books, Enron), and large-scale financial networks (Dgraph), ensuring the reliability and generalizability of the experimental results.

**Weaknesses:**

1、Although the paper mentions that graph encoders (e.g., GCN in Graph AE) may limit the model's ability to represent complex graph structures and relationships, it only briefly notes this issue without analyzing how this limitation impacts DiffGAD’s performance in specific experiments or real-world scenarios. For example, when handling highly heterogeneous or dynamically changing graph structures, the encoder may fail to accurately capture critical information, potentially leading to reduced anomaly detection accuracy.

2、For the key hyperparameter λ in the model, simply showing the impact of different values on performance through experiments is insufficient. There is a lack of theoretical justification explaining why the optimal value differs across datasets and the intrinsic connection between these values and data characteristics.

3、In the introduction, the logical connection between the research motivation and the subsequent presentation of DiffGAD's innovations could be clarified. It would be helpful to explicitly highlight the specific limitations of traditional methods in capturing discriminative content and how DiffGAD’s unique design (such as the generative paradigm guided by discriminative content and content preservation strategy) directly addresses these issues.

**Questions:**

The paper mentions that graph encoders (such as GCN in Graph AE) might limit the model’s ability to represent complex graph structures and relationships, but it does not detail how this limitation manifests in practical applications or to what extent it impacts DiffGAD’s performance. For instance, when processing graphs with highly heterogeneous node attributes or complex topologies (e.g., multi-layer nested structures, frequently dynamic graphs), would the current encoder lead to critical information loss? How does this information loss impact the accurate extraction of discriminative content and, subsequently, anomaly detection accuracy? The authors could quantify this impact by designing targeted experiments. For example, they could construct synthetic graph datasets with varying levels of heterogeneity and topological complexity to compare DiffGAD’s performance when using the current encoder versus a hypothetical ideal encoder (with stronger representational power). Metrics such as anomaly detection accuracy and recall could provide insight into performance differences.

---

> ### Author Response · Authors · 2024-11-24
> **Response to Reviewer 2tPm**
>
> **Response to Reviewer $\color{green}{\text{2tPm}}$**
>
> We sincerely appreciate your insightful comments and acknowledgment of our contributions. Below we provide the point-to-point responses to address your concerns and clarify the confusion of our proposed method. If you have additional questions, we would be pleased to discuss them with you.
>
> > **Comment 1 + Questions: Experiments with highly heterogeneous and dynamically changing graph structures** - "when handling highly heterogeneous or dynamically changing graph structures..."
>
> Thank you for raising this point! We can fully understand your concerns!
>
> Initially, our focus was not on heterogeneous or dynamically changing graph structures. Our encoder may fail to accurately capture critical information when dealing with such data, potentially leading to reduced anomaly detection accuracy. To investigate whether this is the case, we utilize the heterogeneous graph dataset IMDB proposed in HAN [1]. Since the IMDB dataset does not inherently contain anomaly samples, we follow the approach outlined in BOND [2] to manually inject structural and contextual anomalies into the IMDB dataset with an anomaly ratio of 4.9%. We explore several architectures, specifically GCN in Graph AE, GCN in VGAE, and Graph Transformer in Graph AE. Their individual performances, as well as their performance in conjunction with our DiffGAD, are presented as follows:
>
> **Table 1: The ROC-AUC (%) performance of different encoder architectures and backbones on IMDB, where we show the *avg perf.* ± *the std of perf*.**
>
> |      |    GAE     |    VGAE    | GraphTransformer |
> | :--: | :--------: | :--------: | :--------------: |
> | IMDB | 67.79±0.00 | 42.91±2.70 |    67.84±0.00    |
>
> **Table 2: The ROC-AUC (%) performance of different encoder architectures and backbones with our DiffGAD on IMDB, where we show the *avg perf.* ± *the std of perf*.**
>
> |      | DiffGAD(GAE) | DiffGAD(VGAE) | DiffGAD(GTrans) |
> | :--: | :----------: | :-----------: | :-------------: |
> | IMDB |  68.76±0.00  |  44.55±2.20   |   68.12±0.10    |
>
> We can observe that, the overall performance of VGAE was subpar, indicating that it is not an effective architecture for handling heterogeneous data in our experiments. The use of Graph Transformer alone yields relatively good results, however, when combined with our DiffGAD, our architecture demonstrates a slight advantage. Nevertheless, this is not the optimal outcome. In the future, we can explore the use of different architectures tailored to various types of structures or develop a unified architecture capable of addressing all data types.
>
>
>
> > **Comment 2: Illustrations of hyper-parameter selection** - "For the key hyper-parameter $\lambda$ in the model..."
>
> Thank you for drawing our attention to this important aspect!  Specifically, we find something interesting when calculating the homophily and heterophily of anomaly samples around different datasets, and the results are listed in the following table.
>
> **Table 3: The homophily and heterophily of anomalies on 5 datasets.**
>
> |             | Weibo | Reddit |  Books   |  Disney  | Enron    |
> | :---------: | :---: | :----: | :------: | :------: | -------- |
> |  Homophily  | 0.87  |  0.18  |   0.01   |   0.00   | 1.00     |
> | Heterophily | 0.13  |  0.82  | **0.99** | **1.00** | **0.00** |
>
> And we can draw the following conclusions:
>
> - Larger $\lambda$ works better on the datasets where anomaly samples have large heterophily, and we recommend using $\lambda$  larger than 1 on such data. Larger heterophily for anomaly samples means they are hidden within normal samples, and thus they are harder to detect.
> - Smaller $\lambda$ works well on the datasets where anomaly samples have smaller heterophily, and we recommend using $\lambda$ smaller than 1 on such data. Smaller heterophily for anomaly samples means they are clustered together, and thus they are easier to detect.
>
> > **Comment 3: Logical of the motivation** - "It would be helpful to explicitly highlight..."
>
> Thanks for your kind suggestions! This indeed helps us clarify the writing logic in the Introduction. We will soon make adjustments in the revised version based on your suggestions. Thanks once again for your guidance.
>
>
>
> [1] Xiao Wang, et al. Heterogeneous Graph Attention Network. In WWW 2019.
>
> [2] Kay Liu, et al. BOND: Benchmarking Unsupervised Outlier Node Detection on Static Attributed Graphs. In NeurIPS 2022.

---

> ### Comment · Reviewer_2tPm · 2024-11-26
>
> Thank you for the author's response. The additional discussion has largely addressed my concerns! I truly appreciate it.

---

> > ### Author Response · Authors · 2024-11-26
> > **Heartfelt Thanks to Reviewer 2tPm!**
> >
> > Dear Reviewer $\color{green}{\text{2tPm}}$,
> >
> > We greatly appreciate for keep supporting us! Your positive feedback and constructive suggestions have been instrumental in improving the quality of our work. If you have any additional questions or concerns that we can clarify or address, don't hesitate to contact us.
> >
> > Thank you once again for your valuable time and effort in reviewing our updated submission!
> >
> > Best regards,
> >
> > Authors of 6534.

---

### Official Review · Reviewer_wf4Y · 2024-11-02

**Soundness:** 3
**Presentation:** 3
**Contribution:** 3
**Rating:** 6
**Confidence:** 3

**Summary:**

This article presents DiffGAD, a new method based on Diffusion Models (DM), for dealing with the problem of anomaly detection in graph data. A research framework based on Diffusion Models is constructed, which encodes the graph data into the latent space via an encoder, then adds noise to preserve the general content and samples from unconditional and conditional diffusion models, and finally transforms the reconstructed embedding back into the graph space via a decoder to compute the reconstruction error. Finally, through experiments on multiple real-world datasets, the authors conclude the effectiveness of DiffGAD on graph anomaly detection tasks.

**Strengths:**

1、The method combines graph neural networks (GNNs) with diffusion models, an innovative fusion of techniques that takes advantage of both the strengths of GNNs in graph structural analysis and the sophistication of diffusion models in generative modeling.
2、 DiffGAD uses both conditional and unconditional diffusion models to reconstruct the graph, and this combination improves the model's sensitivity and ability to recognize anomalies.

**Weaknesses:**

1、the experimental part of the dataset settings, the dataset selected in this paper are relatively small feature dimensions, have you considered the use of high-dimensional features of the dataset for the experiment?
2、Comparison methods on although there are based on deep learning, but the latest is 2022, it is recommended to increase the comparison experiments, and the current advanced methods in the field, to show the advantages of the project method.

**Questions:**

1、How do the two DMs achieve concurrent sampling without adding extra parameters?
2. In 3.3, the minimum perturbation is used to change the potential embedding z0 to zt. How is this “minimum perturbation” defined and quantified?

---

> ### Author Response · Authors · 2024-11-24
> **Response to Reviewer wf4Y ---- Part 1/2**
>
> **Response to Reviewer $\color{purple}{\text{wf4Y}}$**
>
> We highly appreciate your invaluable comments and positive feedback on our work, which inspires us to greatly improve our paper. To address your concerns, we present the point-to-point responses as follows. We will carefully revise our paper, taking all your feedback into account.
>
> > **Comment 1: Experiments with high-dimensional features** - "have you considered the use of high-dimensional features of the dataset ...?"
>
> Thank you for raising this point! We fully understand your concerns and we try to solve them from two perspectives.
>
> - **Can our method perform well on datasets with large feature dimensions?**
>
> The feature dimension of the datasets we select is relatively small, with a maximum of 400. To validate the scalability of our method to high-dimensional data, we conducted experiments on two datasets, Cora and Flickr, mentioned in Bond [1]. The details of the datasets and the experimental results are presented below:
>
> **Table 1: Detailed statistics of Cora and Flickr.**
>
> |  | #Nodes | **#Edges** | **#Feat** | **Degree** | **#Outliers** | **Ratio** |
> | :--------: | :----: | :--------: | :-------: | :--------: | :-----------: | :-------: |
> |  Cora [2]  |  2708  |   11060    |   1433    |    4.1     |      138      |   5.1%    |
> | Flickr [3] | 89250  |   933804   |    500    |    10.5    |     4414      |   4.9%    |
>
> **Table 2: Performance comparison (ROC-AUC) among 8 deep algorithms on Cora and Flickr, where we show the *avg perf.* *±* *the std of perf.* of each method.**
>
> | |     Cora     |    Flickr    |
> | :--------: | :----------: | :----------: |
> |   GCNAE    |   70.9±0.0   |   71.6±3.1   |
> |  DOMINANT  |   82.7±5.6   |  78.0±12.0   |
> |    DONE    |   82.4±5.6   | **84.7±2.5** |
> |   AdONE    |   81.5±4.5   |   82.8±3.2   |
> | AnomalyDAE |   83.4±2.3   |   65.6±3.5   |
> |    GAAN    |   74.2±0.9   |   72.4±0.2   |
> |   CONAD    |   78.8±9.6   |   65.1±2.5   |
> |  DiffGAD   | **84.5±0.0** |   83.3±0.0   |
>
> We can observe that our method can also achieve strong performance on high-dimensional feature datasets.
>
> - **If different dimensions of the feature have impacts on our method?**
>
> To explore this perspective, we conduct experiments on Weibo using different latent embedding dimensions, and the results are presented below:
>
> **Table 3: Performance comparison (ROC-AUC) of 7 different dimensions on Weibo, where we show the *avg perf.* *±* *the std of perf.* of each method.**
>
> | |    8     |    32    |  **64**  |   128    |   256    |   400    |   512    |
> | :---------: | :------: | :------: | :------: | :------: | :------: | :------: | :------: |
> | **DiffGAD** | 93.9±0.5 | 92.7±0.1 | 92.8±0.1 | 93.4±0.3 | 93.0±0.1 | 92.8±0.5 | 92.6±0.4 |
>
> We can observe that our method can effectively leverage feature information across different dimensions, and the dimensions of the feature has minimal impact on our experimental results.
>
>
>
> > **Comment 2: Comparison with more methods** - "it is recommended to increase the comparison experiments..."
>
> Thanks for your invaluable comments! Folloing your suggestion, we have **conducted additional experiments** on GAD-EBM [4], GDSS_Rec and GDSS_Energy [5] on 5 datasets. We will add the latest baselines in our revised submission!
>
> **Table 4: ROC-AUC (%) comparison with latestest baselines on 5 datasets, where we show the *avg perf.* ± *the std of perf*. TLE indicates that the method exceeded the time limit of 24 hours.**
>
> |  |    Weibo     |  Reddit  |  Disney  |  Books   |  Enron   |   Avg |  Dgraph  |
> | :---------: | :----------: | :----------: | :-----------: | :----------: | :----------: | :----------: | :----------: |
> |   GAD-EBM   |   84.5±8.5   |   53.8±5.7   |   57.4±15.1   |   62.9±0.7   | **79.7±2.4** |   67.7±6.5   | **60.3±2.5** |
> |  GDSS_Rec   |  74.5±12.6   |   44.5±0.4   | **65.0±11.2** |   57.1±2.8   |   44.0±4.0   |   57.0±6.2   |     TLE      |
> | GDSS_Energy |  51.9±11.1   |   55.1±0.8   |   58.4±5.9    |   52.7±3.0   |   36.5±5.6   |   50.9±5.3   |     TLE      |
> |   DiffGAD   | **93.4±0.3** | **56.3±0.1** |   54.5±0.2    | **66.4±1.8** |   71.6±7.0   | **68.4±1.9** |   52.4±0.0   |
>
> We can observe that:
>
> - DiffGAD consistently exhibits superior performance compared to an energy-based method and score-based baseline across varying datasets in terms of ROC-AUC metric. Such results underscore the robustness of DiffGAD.
> - GAD-EBM slightly edges out DiffGAD on Enron and Dgraph, a potential explanation lies in that GAD-EBM takes ego-subgraph presentation into consideration with an energy-based model, which explicitly utilizes more structural information.
> - GDSS_Rec and GDSS_Energy directly apply the score-based diffusion model to graph features and structures to calculate the reconstruction errors, which show the best result on small dataset Disney but poor results on larger ones. We attribute this to the direct utilization of node features and structural information by the diffusion model.

---

> ### Author Response · Authors · 2024-11-24
> **Response to Reviewer wf4Y ---- Part 2/2**
>
> > **Question 1: Illustration of the sampling process** - "How do the two DMs achieve concurrent sampling without adding extra parameters?"
>
> Thank you for raising this point!
>
> So sorry for the confusion! Actually, this concurrent sampling is inspired by classifier-free guidance (CFG) [6], which aims to train 2 separate DMs to achieve concurrent sampling. But with the high computational cost, it trains one model with a well-curated condition dropping ratio for simplicity. In our scenario, we notice that our diffusion is computationally efficient, it can support us to train 2 separate DMs and achieve a good performance without adding many parameters, as discussed in Section 5 of our time and computational analysis. We will modify this point in our revised version.
>
>
>
> > **Question 2: Illustration of the perturbutation** - "How is this “minimum perturbation” defined and quantified?"
>
> Thanks for your kind suggestions, and sorry for such confusion. We intend to express "Small amount of" in this scenario, and we will modify this in the paper.
>
> Moreover, our quantification can be described as follows:
>
> Specifically, to fine-grained investigate the effectiveness of noise scales, following the work in [7], we categorize the noises into 500 different scales, and we add noises at each scale by the following equation:
>
> $z_{t} = (t/T) z_{0} + ((T-t)/T) \epsilon$, where $\epsilon$ is a random noise, and $\epsilon \in \mathbb{N}(0,1)$. Specifically, when we set  $t=1$, it implies that we add minimum noises to the original feature, and when $t=500$, we sample from the random noises.
>
> Thank you again for carefully pointing this out! We will supplement this in the method of our paper.
>
>
>
> [1] Kay Liu, et al. BOND: benchmarking unsupervised outlier node detection on static attributed graphs. In NeurIPS 2022.
>
> [2] Prithviraj Sen et, al. Collective classification in network data. In AI magazine 2008.
>
> [3] Hanqing Zeng, et al. Graphsaint: Graph sampling based inductive learning method. In ICLR 2020.
>
> [4] Amit Roy, et al. GAD-EBM: Graph Anomaly Detection using Energy-Based Models. In NeurIPS 2023.
>
> [5] Dmitrii Gavrilev, et al. Anomaly Detection in Networks via Score-Based Generative Models. In ICML 2023.
>
> [6] Jonathan Ho, et al. Classifier-Free Diffusion Guidance. In NeurIPS 2021.
>
> [7] Jonathan Ho, et al. Denoising Diffusion Probabilistic Models. In NeurIPS 2020.

---

> ### Author Response · Authors · 2024-12-02
> **Greatly appreciate your support!**
>
> Dear Reviewer $\color{purple}{\text{wf4Y}}$,
>
> We are extremely grateful for the recognition you have shown toward our work and insightful suggestions to take latest methods for comparison. We also try our best to address your concerns about the high-dimensional features, DMs, and some confusion about $t$.
>
> As the discussion phase comes to a close, **if our response has solved most of your concerns, we humbly ask if you might consider raising your score slightly**.
>
> If you believe there are still areas in our paper that could be further improved, we would be more than happy to engage in any additional discussion to address any remaining concerns, particularly as the rebuttal period is about to conclude in just a few hours.
>
> Thank you once again for your thoughtful engagement! Your support at this stage is immensely important to us!
>
> Best regards,
>
> Authors of Paper 6534

---

### Official Review · Reviewer_fVC2 · 2024-11-02

**Soundness:** 3
**Presentation:** 3
**Contribution:** 3
**Rating:** 8
**Confidence:** 3

**Summary:**

The paper introduces DiffGAD, a diffusion based model designed for graph anomaly detection (GAD). It employs a latent space learning paradigm and incorporates discriminative content to enhance profiency. The authors present experimental results on six large real world datasets to demonstrate model performance. Code is made available as well.

**Strengths:**

1. The use of diffusion models for GAD is an innovative application
2. A generous number of meaningful datasets are used for experimentation
3. The figures used in the paper manage to illustrate the problem and approach well.

**Weaknesses:**

1. A specific graph autoencoder (AE) is used by the model. This may limit the adaptability of the model. (See Q1)
2. Hyperparameters like λ affect the performance of DiffGAD. If this turns in to an "art" to get the best performance, then it may be problematic for real world scenarios. Clarification from the authors about some guidelines to selecting λ could be helpful.

**Questions:**

1. Did the authors explore any alternatives to the currently used AE?
2. Follow-up: a less comlex alternative could give an efficiency boost at some performance cost. Would be interesting to see. (not required for acceptance)
3. Are there any particular type of datasets that are more "conducive" to DiffGAD than others?

---

> ### Author Response · Authors · 2024-11-24
> **Response to Reviewer fVC2 ---- Part 1/2**
>
> **Response to Reviewer $\color{blue}{\text{fVC2}}$**
>
> Thanks so much for your time and positive feedback! To address your concerns, we have detailed our responses point-to-point below.
>
> > **Comment 1 + Question 1: Adaptability of the model** - "A specific graph autoencoder(AE) is used...", "... any alternative ...?"
>
> Thanks for your thoughtful comment and question! It is indeed necessary to explore whether the autoencoder (AE) currently in use is the most effective option and to investigate potential alternative architectures. Here we try 4 alternative architectures of AE: MLP, VAE, VGAE, and GTrans respectively.
>
> **Table 1: The ROC-AUC (%) performance of different AE architectures, where we show the *avg perf.* ± *the std of perf*.**
>
> |  Backbone  |  Weibo  |  Reddit  | Disney   | Books  |   Enron |   Avg  |
> | :--------: | :----------: | :----------: | :----------: | :----------: | :----------: | :----------: |
> | MLP |   85.5±0.3   |   50.6±0.0   | **48.1±0.1** |   49.3±5.9   |  41.3±14.1   |   55.0±4.1   |
> |  VAE   |   84.4±0.2   |   50.6±0.0   |   48.0±0.1   |   50.5±5.4   |   42.9±8.2   |   55.3±2.8   |
> |  VGAE  | **92.5±0.0** |   55.8±0.1   |   47.8±3.3   |   48.4±4.2   |   56.8±0.7   |   60.3±1.7   |
> | GTrans |   90.9±0.4   |   56.1±0.0   |   31.0±7.0   |  32.5±12.3   |   55.5±0.6   |   53.2±4.1   |
> | GAE (Ours) |   92.0±0.1   | **56.1±0.0** |   40.3±6.7   | **59.1±2.5** | **59.1±1.6** | **61.3±2.2** |
>
> **Table 2: The ROC-AUC (%) performance of different AE with our DiffGAD, where we show the *avg perf.* ± *the std of perf*.**
>
> |  Method | Weibo | Reddit | Disney| Books | Enron | Avg |
> | :-------------: | :----------: | :----------: | :----------: | :----------: | ------------ | :----------: |
> |  DiffGAD(MLP)   |   90.0±1.1   |   50.6±0.0   |   49.1±0.2   |   50.0±0.0   | **81.9±5.2** |   64.3±1.3   |
> |  DiffGAD(VAE)   |   90.5±0.6   |   50.6±0.0   |   48.5±0.2   | **67.1±2.9** | 66.0±1.8     |   64.5±1.1   |
> |  DiffGAD(VGAE)  |   92.5±0.0   |   56.1±0.0   |   52.1±3.2   |   51.0±1.1   | 57.8±0.2     |   61.9±0.5   |
> | DiffGAD(GTrans) |   93.0±0.7   |   56.2±0.1   |   51.4±0.6   |   51.5±0.5   | 65.7±4.9     |   63.6±1.4   |
> |  DiffGAD(GAE)   | **93.4±0.3** | **56.3±0.1** | **54.5±0.2** |   66.4±1.8   | 71.6±7.0     | **68.4±1.9** |
>
> For a more comprehensive explanation, we not only test the anomaly detection performance of different AEs but also compare the performance of AEs with our DiffGAD to prove the scalability and generalization ability of DiffGAD.
>
> We can observe that:
>
> - GAE and DiffGAD(GAE) consistently exhibit superior performance compared to different encoder & decoder architectures across different datasets in terms of ROC-AUC. Such results underscore the robustness of GAE and the generalization ability of DiffGAD.
> - The significant performance gains of DiffGAD(MLP) on the Enron dataset, and DiffGAD(VAE) on the Books dataset demonstrate that the non-graph based method may be more suited for these datasets, and also show that the data characteristic plays a key role in determining the effectiveness of the model.
>
> - The architecture of Graph Transformer may be somewhat excessive for this task, as it exhibits a relatively high complexity, but without demonstrating any improvement in performance.
>
> To sum up: we believe that **The greatest truths are the simplest**, and a simple architecture, such as GAE, is enough for latent space construction over current datasets, which preserves sufficient discriminative information, and further works with this latent space can boost the detection performance.  Moving forward, we will investigate various AE architectures in more complex applications to uncover additional insights. Thank you!
>
> > **Comment 2: Guidelines for hyper-parameter selection.** - "Clarification from the authors about some guidelines to selecting $\lambda$. could be helpful.."
>
> Thank you for drawing our attention to this important aspect!  Specifically, we calculate the Homophily and Heterophily of anomaly samples around different datasets, and the results are listed in the following table.
>
> **Table 3: The homophily and heterophily of anomalies on 5 datasets.**
>
> | | Weibo | Reddit |  Books   |  Disney  | Enron |
> | :---------: | :---: | :----: | :------: | :------: | -------- |
> |  Homophily  | 0.87  |  0.18 |   0.01 |  0.00   | 1.00  |
> | Heterophily | 0.13  |  0.82  | **0.99** | **1.00** | **0.00** |
>
> And we can draw the following conclusions:
>
> - Larger $\lambda$ works better on the datasets where anomaly samples have larger heterophily, and we recommend using $\lambda$  larger than 1 on such data. Larger heterophily for anomaly samples means they are hidden within normal samples, and thus they are harder to detect.
> - Smaller $\lambda$ works well on the datasets where anomaly samples have smaller heterophily, and we recommend using $\lambda$ smaller than 1 on such data. Smaller heterophily for anomaly samples means they are clustered together, and thus they are easier to detect.

---

> ### Author Response · Authors · 2024-11-24
> **Response to Reviewer fVC2 ---- Part 2/2**
>
> > **Question 2: More Simpler Architecture.** - "A less complex alternative could give.."
>
> Thank you for raising this point! The trade-off of both efficiency and performance is truly worth exploring.
>
>  For simpler architecture, we try MLP as the backbone of AE to explore its ability in anomaly detection, it has no need to handle the graph structure and thus has lower computational complexity. Though MLP have advantages in terms of time and computational complexity, their performance has not been particularly impressive. The notable results on the Disney dataset **(48.1±0.1)** in **Table 1** lies in that, for small graphs in terms of 'Nodes', 'Edges', and 'Feat' (see Table 5 of our paper), it could be difficult for the deep architectures to encode the inlier distribution well and it could also possibly lead to overfitting issues. In this way, we might adopt simpler architecture like MLP to simultaneously achieve minimal time consumption while maintaining satisfactory performance.
>
> Besides, the best result on Enron **(81.9±5.2)** in **Table 2** also indicates that the graph structure can lead to a loss of discriminative information, as MLPs rely solely on node features as input.
>
> To summarize, following your suggestions, we believe that in the future, it's a good idea to flexibly adopt different AE architectures across various datasets depending on specific requirements (such as when there are high demands for time complexity).
>
>
>
> > **Question 3: Dataset Exploration.** - "Are there any particular type of datasets.."
>
> Thanks for your insightful questions!
>
> This has prompted us to reflect on our approach. From the perspective of our model's motivation, it is better equipped to capture discriminative content. For instance, in the context of fraud detection, fraudsters often camouflage themselves as normal users to bypass the anti-fraud systems and disperse disinformation or reap end-users' privacy. By camouflaging (such as interacting with normal users or regularly posting seemingly benign comments), fraudsters can exhibit numerous shared common content and behavioral patterns with normal users. Therefore, effectively extracting discriminative information that distinguishes between normal and abnormal samples becomes particularly crucial. We believe that our approach can achieve significant results on such applications.

---

> > ### Comment · Reviewer_fVC2 · 2024-11-25
> >
> > I thank the authors for their detailed and thorough response! I appreciate it

---

> > > ### Author Response · Authors · 2024-11-25
> > > **Greatly appreciate your support!**
> > >
> > > Dear Reviewer $\color{blue}{\text{fVC2}}$,
> > >
> > > Thank you for your kind feedback and for taking the time to review our updated work! We are grateful for your recognition, it means a lot to us and inspires us to continue improving.
> > >
> > > We are fully committed to advancing the field of Graph Anomaly Detection and contributing meaningfully to the community. Your feedback and support are invaluable to us in achieving this goal.
> > >
> > > Thank you again for your thoughtful comments and encouragement. We genuinely appreciate your support.
> > >
> > > Best regards,
> > >
> > > Authors of Paper 6534

---

### Official Review · Reviewer_pSKo · 2024-11-04

**Soundness:** 3
**Presentation:** 3
**Contribution:** 2
**Rating:** 6
**Confidence:** 3

**Summary:**

The paper proposes a Diffusion-based Graph Anomaly Detector for unsupervised identifying abnormal entities in networks. The contributions can be summarized as follows: 1: The author made the first attempt to transfer the diffusion models to the graph anomaly diffusion tasks. The model consists of an auto-encoder framework and two diffusion models to conduct unconditional diffusion and conditional diffusion. 2: To guide the training of diffusion models in latent space, the authors propose to use a discriminative content-guided generation paradigm to distill the discriminative content in latent space; and a content-preservation strategy to enhance the confidence of the guidance process. 3: The authors conduct experiments on 7 real-world datasets and make comparisons with 13 baseline methods.

**Strengths:**

- The first attempt to transfer the generative diffusion models to the GAD task is a great try, bringing new perspectives to the anomaly detection community.
- The idea of distilling the discriminative content based on a linear combination of the two different diffusion models is interesting and easy to follow.

**Weaknesses:**

- Some parts of the paper need further explanation. For example, in the introduction, the authors mention that some researchers utilize encoders to map graph data into a latent space, but there is a lack of essential discussion about why they are doing that.
- The motivation behind some of the model designs is ambiguous. For instance, in the selection of the encoder and decoder, the conditions that a good encoder and decoder should satisfy are unclear. Instead of directly utilizing the GAE framework, it seems more important to discuss the criteria for selecting a good encoder and decoder and add related experiments to support your claims. I did not see any discussions related to that.

**Questions:**

- Why did you choose to use GAE as the encoder and decoder instead of considering other models, such as VGAE, graph transformers, etc.? Is it possible to use these models as the encoder and decoder?
- What are the conditions that a good latent space should satisfy? Or, what constitutes a good latent space for performing the diffusion process to conduct anomaly detection tasks?
- Regarding the training process, it looks like you first train a satisfactory encoder and decoder, then fix the parameters, and train the diffusion models. Why did you not choose to construct a global loss and train the GAE and diffusion models together?

---

> ### Author Response · Authors · 2024-11-24
> **Response to Reviewer pSKo ---- Part 1/4**
>
> **Response to Reviewer $\color{red}{\text{pSKo}}$**
>
> We highly appreciate your insightful comments, which help us a lot to better scrutiny and polish our work! We notice that all your comments and questions are focused on the **latent space**. for clear demonstration, we will organize them into the following three aspects:
>
> > **1. Why is mapping graph data into a latent space necessary? ---- Comment 1**
>
> Mapping the graph into latent space has 2 key points, the first is reconstruction error as anomaly scores, and the next is encoder & decoder architecture, details are as follows.
>
> - **Reconstruction Errors**
>
>  A straightforward way to perform anomaly detection in attributed graph networks is to assume that some properties of anomalies are known in advance [1]. In many cases, this assumption might not be true, especially under the unsupervised learning paradigm. Therefore, it is beneficial and desirable to explore and spot anomalies in a general sense without prior knowledge.
>
> As suggested by [2], the disparity between the original data and the estimated data (i.e., **reconstruction errors**) is a strong indicator to show the abnormality of instances in a dataset and this is also the paradigm for unsupervised graph anomaly detection. Specifically, it hypothesizes that anomaly samples are hard to reconstruct and regards the data instances with large reconstruction errors as anomalies.
>
> - **Encoder & Decoder**
>
> The structure of the encoder and decoder are natural tools for calculating the reconstruction errors, with their strong representation ability. Pioneering work [3] applies such architecture to anomaly detection and performs brilliant effects. Inspired by this, researchers [4, 5, 6] from the graph community design GNNs-based AEs to capture the node attributes and graph structure into a unified latent space and demonstrate significant performance gains than directly reconstructing in the graph domain [1], where these significant performance gains demonstrate that latent space can effectively handles both graph node and structure information.
>
> Thanks again for the thoughtful suggestion. It makes perfect sense, we will polish this soon in our revised version.

---

> ### Author Response · Authors · 2024-11-24
> **Response to Reviewer pSKo ---- Part 2/4**
>
> > **2. What is a good latent space/ encoder & decoder? ---- Comment 2 + Question 1 & 2**
>
> Thank you for drawing our attention to this important aspect!
>
> In this paper, We hypothesize that a well-constructed latent space by encoder & decoder (denoted as general content in paper) should encode enough discriminative content,  and our discriminative distillation aims to mine such content. In our paper, experimental results show the effectiveness of discriminative distillation but do not list the importance of latent space construction. To answer this, we design 4 different encoder & decoder architectures, including VGAE, MLP, VAE, Graph Transformer. The results are listed in the table.
>
> **Table 1: The ROC-AUC (%) performance of different encoder & decoder architectures, where we show the *avg perf.* ± *the std of perf*.**
>
> |  Backbone  |    Weibo     |    Reddit    |    Disney    |    Books     |    Enron     |     Avg      |
> | :--------: | :----------: | :----------: | :----------: | :----------: | :----------: | :----------: |
> |    MLP     |   85.5±0.3   |   50.6±0.0   | **48.1±0.1** |   49.3±5.9   |  41.3±14.1   |   55.0±4.1   |
> |    VAE     |   84.4±0.2   |   50.6±0.0   |   48.0±0.1   |   50.5±5.4   |   42.9±8.2   |   55.3±2.8   |
> |    VGAE    | **92.5±0.0** |   55.8±0.1   |   47.8±3.3   |   48.4±4.2   |   56.8±0.7   |   60.3±1.7   |
> |   GTrans   |   90.9±0.4   | 56.1 *±* 0.0 |   31.0±7.0   |  32.5±12.3   |   55.5±0.6   |   53.2±4.1   |
> | GAE (Ours) |   92.0±0.1   | **56.1±0.0** |   40.3±6.7   | **59.1±2.5** | **59.1±1.6** | **61.3±2.2** |
>
> **Table 2: The ROC-AUC (%) performance of different encoder & decoder architectures with our DiffGAD, where we show the *avg perf.* ± *the std of perf*.**
>
> |     Method      |    Weibo     |    Reddit    |    Disney    |    Books     | Enron        |     Avg      |
> | :-------------: | :----------: | :----------: | :----------: | :----------: | ------------ | :----------: |
> |  DiffGAD(MLP)   |   90.0±1.1   |   50.6±0.0   |   49.1±0.2   |   50.0±0.0   | **81.9±5.2** |   64.3±1.3   |
> |  DiffGAD(VAE)   |   90.5±0.6   |   50.6±0.0   |   48.5±0.2   | **67.1±2.9** | 66.0±1.8     |   64.5±1.1   |
> |  DiffGAD(VGAE)  |   92.5±0.0   |   56.1±0.0   |   52.1±3.2   |   51.0±1.1   | 57.8±0.2     |   61.9±0.5   |
> | DiffGAD(GTrans) |   93.0±0.7   |   56.2±0.1   |   51.4±0.6   |   51.5±0.5   | 65.7±4.9     |   63.6±1.4   |
> |  DiffGAD(GAE)   | **93.4±0.3** | **56.3±0.1** | **54.5±0.2** |   66.4±1.8   | 71.6±7.0     | **68.4±1.9** |
>
> We observe that:
>
> - GAE and DiffGAD(GAE) consistently exhibit superior performance compared to different encoder & decoder architectures across different datasets in terms of ROC-AUC. Such results underscore the robustness of GAE and the generalization ability of DiffGAD.
> - Without encoding graph structure, the DiffGAD(MLP) achieves notable results on the Enron dataset, this indicates that the graph structure can lead to a loss of discriminative information, as MLPs rely solely on node features as input.
> - The different performance of GAE, VAE, and VGAE reflects the impact of different encoder and decoder architectures. However, the architecture of Graph Transformer may be somewhat excessive for this task, as it exhibits a relatively high complexity, but without demonstrating any performance improvement.
>
> To sum up: we believe that **"The greatest truths are the simplest"**, and a simple architecture (i.e. GAE) is enough for latent space construction over current datasets, which preserves sufficient discriminative information, and further works with this latent space can boost the detection performance.
>
> Thanks again for the thoughtful suggestion. It makes perfect sense, we will add this in the appendix of our revised version.

---

> ### Author Response · Authors · 2024-11-24
> **Response to Reviewer pSKo ---- Part 3/4**
>
> > **3. How to construct a good latent space? ---- Question 3**
>
> Thanks for your insightful suggestions! This is an important question!
>
> In this paper, we first construct a latent space and then perform discriminative distillation on the well-constructed space. As you discussed, we can jointly learn the DM and AE, which concretely construct a latent space and mine the discriminative content, and the results are listed in the table.
>
> **Table 3: The ROC-AUC (%) performance comparison of separate and joint training AE & DM based on MLP, where we show the *avg perf.* ± *the std of perf*.**
>
> |              |    Weibo     |    Reddit    |    Disney    |    Books     |    Enron     |     Avg      |
> | :----------: | :----------: | :----------: | :----------: | :----------: | :----------: | :----------: |
> |   AE(MLP)    |   85.5±0.3   |   50.6±0.0   |   48.1±0.1   |   49.3±5.9   |  41.3±14.1   |   55.0±4.1   |
> |  Joint(MLP)  |   83.5±0.4   |   50.6±0.0   |   48.0±0.1   |   48.8±6.5   |   42.9±6.5   |   54.8±2.7   |
> | DiffGAD(MLP) | **90.0±1.1** | **50.6±0.0** | **49.1±0.2** | **50.0±0.0** | **81.9±5.2** | **64.3±1.3** |
>
> **Table 4: The ROC-AUC (%) performance comparison of separate and joint training AE & DM based on VAE, where we show the *avg perf.* ± *the std of perf*.**
>
> |              |    Weibo     |    Reddit    |    Disney    |    Books     |    Enron     |     Avg      |
> | :----------: | :----------: | :----------: | :----------: | :----------: | :----------: | :----------: |
> |     VAE      |   84.4±0.2   |   50.6±0.0   |   48.0±0.1   |   50.5±5.4   |   42.9±8.2   |   55.3±2.8   |
> |  Joint(VAE)  |   83.3±0.5   |   50.6±0.0   |   48.1±0.1   |   51.0±5.8   |   43.7±8.8   |   55.3±3.0   |
> | DiffGAD(VAE) | **90.5±0.6** | **50.6±0.0** | **48.5±0.2** | **67.1±2.9** | **66.0±1.8** | **64.5±1.1** |
>
> **Table 5: The ROC-AUC (%) performance comparison of separate and joint training AE & DM based on VGAE, where we show the *avg perf.* ± *the std of perf*.**
>
> |               |    Weibo     |    Reddit    |    Disney    |    Books     |    Enron     |     Avg      |
> | :-----------: | :----------: | :----------: | :----------: | :----------: | :----------: | :----------: |
> |     VGAE      |   92.5±0.0   |   55.8±0.1   |   47.8±3.3   |   48.4±4.2   |   56.8±0.7   |   60.3±1.7   |
> |  Joint(VGAE)  |   92.5±0.0   |   55.9±0.1   |   49.8±1.2   | **57.8±3.4** |   43.9±0.0   |   60.0±0.9   |
> | DiffGAD(VGAE) | **92.5±0.0** | **56.1±0.0** | **52.1±3.2** |   51.0±1.1   | **57.8±0.2** | **61.9±0.5** |
>
> **Table 6: The ROC-AUC (%) performance comparison of separate and joint training AE & DM based on Graph Transformer, where we show the *avg perf.* ± *the std of perf*.**
>
> |                 |    Weibo     |    Reddit    |    Disney    |    Books     | Enron        |     Avg      |
> | :-------------: | :----------: | :----------: | :----------: | :----------: | ------------ | :----------: |
> |     GTrans      |   90.9±0.4   | 56.1 *±* 0.0 |   31.0±7.0   |  32.5±12.3   | 55.5±0.6     |   53.2±4.1   |
> |  Joint(GTrans)  |   91.8±0.2   |   56.1±0.0   |   48.0±0.0   | **55.0±3.7** | 58.7±1.9     |   61.9±1.2   |
> | DiffGAD(GTrans) | **93.0±0.7** | **56.2±0.1** | **51.4±0.6** |   51.5±0.5   | **65.7±4.9** | **63.6±1.4** |
>
> **Table 7: The ROC-AUC (%) performance comparison of separate and joint training AE & DM based on GAE, where we show the *avg perf.* ± *the std of perf*.**
>
> |              |    Weibo     |    Reddit    |    Disney    |    Books     |    Enron     |     Avg      |
> | :----------: | :----------: | :----------: | :----------: | :----------: | :----------: | :----------: |
> |  GAE(Ours)   |   92.0±0.1   |   56.1±0.0   |   40.3±6.7   |   59.1±2.5   |   59.1±1.6   |   61.3±2.2   |
> |  Joint(GAE)  |   92.5±0.0   |   56.1±0.0   |   49.5±0.6   |   56.5±3.9   |   58.2±1.7   |   62.6±1.2   |
> | DiffGAD(GAE) | **93.4±0.3** | **56.3±0.1** | **54.5±0.2** | **66.4±1.8** | **71.6±7.0** | **68.4±1.9** |
>
> We can observe that:
>
> - DiffGAD outperforms joint training across various encoder and decoder architectures. A potential explanation for this is that diffusion models are capable of capturing discriminative content in well-defined data distribution [4, 5, 6]. During joint training, the latent space generated by the AE is in constant flux, while the diffusion model is highly sensitive to distributional changes, leading to challenges related to training instability. This issue is particularly pronounced in smaller datasets, such as Disney, Books, and Enron, where the prior information is inherently limited, resulting in a significant decline in performance for joint training.
>
> Thanks again for valuable suggestions, your suggestions light up our focus on the great importance on latent space, and we will further explore this in our future work.

---

> ### Author Response · Authors · 2024-11-24
> **Response to Reviewer pSKo ---- Part 4/4**
>
> [1] Jundong Li, et al. Radar: Residual analysis for anomaly detection in attributed networks. In IJCAI 2017.
>
> [2] Hanghang Tong, et al. Non-negative residual matrix factorization with application to graph anomaly detection. In SDM 2011.
>
> [3] Chong Zhou, et al. Anomaly detection with robust deep autoencoders. In KDD 2017.
>
> [4] Kaize Ding et al. Deep anomaly detection on attributed networks. In SDM 2019.
>
> [5] Haoyi Fan, et al. Anomalydae: Dual autoencoder for anomaly detection on attributed networks. In IEEE ICASSP 2020.
>
> [6] Zhiming Xu, et al. Contrastive attributed network anomaly detection with data augmentation. In PAKDD 2022.

---

> ### Comment · Reviewer_pSKo · 2024-11-27
>
> The authors have addressed most of my concerns. As such, I have raised my score accordingly.

---

> > ### Author Response · Authors · 2024-11-28
> > **Heartfelt Gratitude to pSKo for increasing the score!**
> >
> > Dear Reviewer $\color{red}{\text{pSKo}}$,
> >
> > We greatly appreciate your recognition of our efforts, your support is our greatest motivation! If you have any additional questions, please feel free to contact us. We would be more than happy to engage in further discussions at your convenience.
> >
> > Thanks again for your thoughtful review and your decision to increase the score!
> >
> > Best regards,
> >
> > Authors of 6534

---

### Author Response · Authors · 2024-12-04

Dear Area Chair and Reviewers,

We truly appreciate your efforts and valuable suggestions in reviewing our paper. We are glad that most reviewers reached a positive consensus on our work's motivation, presentation, novelty, and experimental effectiveness. Since we received a decent number of reviews, we provide a summary on the reviewers’ major feedback and our corresponding actions:

> **Strengths**

- **Motivation**:
  - "The first attempt...is a great try, bringing new perspectives to the community."  (Reviewer $\color{red}{\text{pSKo}}$)
  - "This combination improves the model's sensitivity and ability to recognize anomalies." (Reviewer $\color{purple}{\text{wf4Y}}$)
  - "Offering a fresh approach to address the challenge." (Reviewer $\color{green}{\text{2tPm}}$)
  - "This paper presents a clear motivation." (Reviewer $\color{orange}{\text{epRR}}$)
- **Novelty**:
  - "The idea is interesting and easy to follow." (Reviewer $\color{red}{\text{pSKo}}$)
  - "The use of diffusion models for GAD is an innovative application." (Reviewer $\color{blue}{\text{fVC2}}$)
  - "An innovative fusion of techniques." (Reviewer $\color{purple}{\text{wf4Y}}$)
  - "Innovatively adapt diffusion models from generative tasks to the field of graph anomaly detection." (Reviewer $\color{green}{\text{2tPm}}$)
- **Presentation**:
  - "The figures used in the paper manage to illustrate the problem and approach well." (Reviewer $\color{blue}{\text{fVC2}}$)
  - "The logical structure of the paper is clear." (Reviewer $\color{orange}{\text{epRR}}$)
- **Promising results**:
  - "Ensuring the reliability and generalizability of the experimental results." (Reviewer $\color{green}{\text{2tPm}}$)

> **Responses**

- **Reviewer  $\color{red}{\text{pSKo}}$**:

  The concerns are focused on latent space construction, we address this from the following 2 aspects:

  - Further explanation: We explain why some researchers utilize encoders to map graph data into a latent space (Introduction). We also explain why we not choose to construct a global loss and train the GAE and diffusion models together from the perspective of theoretical and empirical analysis.
  - More experimental exploration: We explore the impact brought from the different encoder & decoder in our method and discuss the criteria for selecting a good encoder & decoder/latent space. (Appendix C.5)

- **Reviewer $\color{blue}{\text{fVC2}}$**:

  - Adaptability of the model: We try 4 alternative AE architectures and explore their applicable scenarios. (Appendix C.5)
  - Guidelines for selecting hyper-parameter $\lambda$: We explore the relationship between $\lambda$ and the heterophily of anomalies to offer better guidance. (Appendix C.6)

- **Reviewer $\color{purple}{\text{wf4Y}}$**:

  - Further illustration: We explain the sampling process and the perturbation and we also modify the expression. (Section 3.1 & 3.3)
  - More experimental evaluation: We compare DiffGAD with 2 latest methods.

- **Reviewer $\color{green}{\text{2tPm}}$**:

  - Explore the limitation: We conduct more evaluations on heterogeneous graphs.
  - Clarify the motivation: We modify the logic in the Introduction according to your suggestion.
  - Guidelines for selecting hyper-parameter $\lambda$: We explore the relationship between $\lambda$ and the heterophily of anomalies. (Appendix C.6)

- **Reviewer $\color{orange}{\text{epRR}}$**:

  The concerns are focused on our experiments, we address this from the following 3 aspects:

  - More experimental validation: We further explain the visualization results and improve the visualization expression (Appendix B). We also further explain why add noise to the features.
  - More experimental evaluation: We compare DiffGAD with 2 latest methods.
  - More experimental analysis: We further analyze the performance of DiffGAD according to your suggestion. (Appendix C.1)

We are pleased that most of the concerns have been clarified, and reviewers expressed satisfaction with our paper and acknowledged our efforts. Reviewer  $\color{red}{\text{pSKo}}$ and $\color{orange}{\text{epRR}}$ consider that we have largely addressed their concerns and **raise their scores from 5 to 6 and from 3 to 5, respectively**! Reviewer  $\color{blue}{\text{fVC2}}$ and $\color{green}{\text{2tPm}}$ **appreciate our efforts  for the detailed and thorough response, and keep supporting us**. Reviewer $\color{purple}{\text{wf4Y}}$ acknowledge our approach and keep supporting us.

Once again, we deeply appreciate the time and expertise you have shared with us. Your encouraging feedback motivates us to continue advancing this work for the broader community, and we are more than happy to add clarifications to address any additional recommendations and reviews from you！

Best regards,

Authors of Paper 6534

---

### Meta-Review · Area_Chair_Sp9L · 2024-12-17

**Metareview:**

This paper proposes a diffusion-based graph anomaly detector (DiffGAD) to identify abnormal entities in networks.  DiffGAD encodes graph data into a latent space via an encoder, adds noise and samples using unconditional and conditional diffusion models, and transformers embeddings back to the graph space via a decoder.  The authors demonstrate DiffGAD's effectiveness compared to baseline deep anomaly detectors on a variety of datasets.

Reviewers leaned positively towards this work, and appreciated the adaptation of diffusion models to the GAD space and the breadth of experiments authors conducted.  There were a few points of concern reviewers raised:

- Clarification around conditions of what a good latent space is for anomaly detection and how this influences the autoencoder choice (pSKo, fVC2, 2tPm)

- Some newer baselines were missed in the initial draft; reviewers addressed some of these concerns, but folding them in and presenting the comparisons with the most recent work is important (wf4Y, epRR)

- Concerns around hyperparameter (\lambda) sensitivity (2tPm, fVC2)

I encourage the authors to fold these into the revision.

**Additional Comments On Reviewer Discussion:**

The authors conducted numerous experiments in response to reviewer concerns.  In response to pSKo, authors explored choices in training the autoencoder and diffusion model jointly vs separately across a number of autoencoder-style backbones.   In response to fVC2,  authors included results for alternatives of autoencoders like MLP, VAE, VGAE, and GTrans, and homophily statistics.  In response to wf4Y, the authors included experiments with high-dimensional features against anomaly detector models, as well as additional baselines like GAD-EBR and GDSS.  In response to 2tPm who asked about dynamic and heterogeneous data, the authors conduced experiments on IMDB.  In response to epRR, the authors conducted experiments on Books dataset as well.  Overall, authors were deeply engaged throughout rebuttal and discussion and were attentive to reviewer feedback.

---

### Decision · Program_Chairs · 2025-01-22

Accept (Poster)